# Cost-Sensitive Self-Training for Optimizing Non-Decomposable Metrics

**Harsh Rangwani**[*1], **Shrinivas Ramasubramanian**[*1], **Sho Takemori**[*2], **Takashi Kato**[2], **Yuhei Umeda**[2], **and R. Venkatesh Babu**[1]

[1]Video Analytics Lab, Indian Institute of Science, Bengaluru, India
[2]Fujitsu Limited, Kanagawa, Japan
harshr@iisc.ac.in, shrinivas.ramasubramanian@gmail.com,
takemori.sho@fujitsu.com, kato.takashi_01@fujitsu.com,
umeda.yuhei@fujitsu.com, venky@iisc.ac.in

## Abstract

Self-training based semi-supervised learning algorithms have enabled the learning of highly accurate deep neural networks, using only a fraction of labeled data. However, the majority of work on self-training has focused on the objective of improving accuracy whereas practical machine learning systems can have complex goals (e.g. maximizing the minimum of recall across classes etc.) that are non-decomposable in nature. In this work, we introduce the Cost-Sensitive Self-Training (CSST) framework which generalizes the self-training-based methods for optimizing non-decomposable metrics. We prove that our framework can better optimize the desired non-decomposable metric utilizing unlabeled data, under similar data distribution assumptions made for the analysis of self-training. Using the proposed CSST framework we obtain practical self-training methods (for both vision and NLP tasks) for optimizing different non-decomposable metrics using deep neural networks. Our results demonstrate that CSST achieves an improvement over the state-of-the-art in majority of the cases across datasets and objectives.

## 1  Introduction

In recent years, semi-supervised learning algorithms are increasingly being used for training deep neural networks [3, 9, 31, 37]. These algorithms lead to accurate models by leveraging the unlabeled data in addition to the limited labeled data present. For example, it's possible to obtain a model with minimal accuracy degradation ($\leq 1\%$) using 5% of labeled data with semi-supervised algorithms compared to supervised models trained using 100% labeled data [31]. Hence, the development of these algorithms has resulted in a vast reduction in the requirement for expensive labeled data.

Self-training is one of the major paradigms for semi-supervised learning. It involves obtaining targets (*e.g.* pseudo-labels) from a network from the unlabeled data, and using them to train the network further. The modern self-training methods also utilize additional regularizers that enforce prediction consistency across input transformations (e.g., adversarial perturbations [18], augmentations [36, 31], etc.) , enabling them to achieve high performance using only a tiny fraction of labeled data. Currently, the enhanced variants of self-training with consistency regularization [40, 25] are among the state-of-the-art (SOTA) methods for semi-supervised learning.

---

[*]Equal Contribution. Code: https://github.com/val-iisc/CostSensitiveSelfTraining

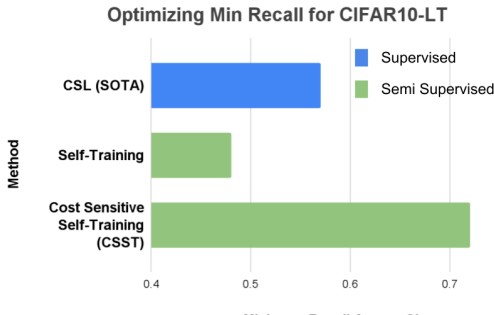

Figure 1: We show a comparison of the SOTA CSL [20] method with the Self-training-based Semi-Supervised methods, for optimizing the minimum recall objective on the CIFAR10-LT dataset. Our proposed CSST framework produces significant gains in the desired metric leveraging additional unlabeled data through our proposed weighted novel consistency regularizer and thresholding mechanism.

Despite the popularity of self-training methods, most of the works [36, 1, 31] have focused on the objective of improving prediction accuracy. However, there are nuanced objectives in real-world based on the application requirements. Examples include minimizing the worst-case recall [19] used for federated learning, classifier coverage for minority classes for ensuring fairness [6], etc. These objectives are complex and cannot be expressed just by using a loss function on the prediction of a single input (i.e., non-decomposable). There has been a considerable effort in optimizing non-decomposable objectives for different supervised machine learning models [20, 30]. However, as supervision can be expensive, in this work we aim to answer the question: *Can we optimize non-decomposable objectives using self-training-based methods developed for semi-supervised learning?*

We first demonstrate that vanilla self-training methods (e.g., FixMatch [31], UDA [36], etc.) can produce unsatisfactory results for non-decomposable metrics (Fig. 1). We then generalize the Cost-Sensitive Loss for Self-Training by introducing a novel weighted consistency regularizer, for a particular non-decomposable metric. Further, for training neural networks we introduce appropriate loss functions and pseudo label selection (thresholding) mechanisms considering the non-decomposable metric we aim to optimize. We also prove that we can achieve better performance on desired non-decomposable metric through our framework utilizing self-training, under similar assumptions on data distributions as made for theoretical analysis of self-training [35]. We demonstrate the practical application by optimizing various non-decomposable metrics by plugging existing methods (*e.g.* FixMatch [31] etc.) into our framework. Our framework leads to a significant average improvement in desired metric of minimizing worst-case recall while maintaining similar accuracy (Fig. 1).

In summary: **a)** we introduce a Cost-Sensitive Self-Training (CSST) framework for optimizing non-decomposable metrics that utilizes unlabeled data in addition to labeled data. (Sec. 4) **b)** we provably demonstrate that our CSST framework can leverage unlabeled data to achieve better performance over baseline on desired non-decomposable metric (Sec. 3) **c)** we show that by combining CSST with self-training frameworks (*e.g.* FixMatch [31], UDA [36] etc.) leads to effective optimization of non-decomposable metrics, resulting in significant improvement over vanilla baselines. (Sec. 5)

## 2 Preliminaries

### 2.1 Non-Decomposable Objectives and Reduction to Cost-Sensitive Learning

We consider the $K$-class classification problem with an instance space $\mathcal{X}$ and the set of labels $\mathcal{Y} = [K]$. The data distribution on $\mathcal{X} \times [K]$ is denoted by $D$. For $i \in [K]$, we denote by $\pi_i$ the class prior $\mathbf{P}(y = i)$. Notations commonly used across paper are in Table 1 present in Appendix. For a classifier $F : \mathcal{X} \rightarrow [K]$, we define confusion matrix $\mathbf{C}[F] \in \mathbb{R}^{K \times K}$ by $C_{ij}[F] = \mathbf{E}_{(x,y) \sim D} [\mathbb{1}(y = i, F(x) = j)]$. Many metrics relevant to classification can be defined as functions of entries of confusion matrices such as class-coverage, recall and accuracy to name a few. We introduce more complex metrics, which are of practical importance in the case of imbalanced distributions [4] (Tab. 1).

Table 1: Metrics defined using entries of a confusion matrix.

| Metric | Definition |
| --- | --- |
| Recall ($\mathrm{rec}_i[F]$) | $\frac{C_{i,i}[F]}{\sum_j C_{i,j}[F]}$ |
| Coverage ($\mathrm{cov}_i[F]$) | $\sum_j C_{j,i}[F]$ |
| Precision ($\mathrm{prec}_i[F]$) | $\frac{C_{i,i}[F]}{\sum_k C_{k,i}[F]}$ |
| Worst Case Recall | $\min_i \frac{C_{i,i}[F]}{\sum_j C_{i,j}[F]}$ |
| Accuracy | $\sum_i C_{i,i}[F]$ |

A classifier often tends to suffer low recalls on tail (minority) classes in such cases. Therefore, one may want to maximize the worst case recall,

$$\max_F \min_{i \in [K]} \text{rec}_i[F].$$

Similarly, on long-tailed datasets, the tail classes suffer from low coverage, lower than their respective priors. An interesting objective in such circumstances is to maximise the mean recall, subject to the coverage being within a given margin.

$$\max_F \frac{1}{K} \sum_{i \in [K]} \text{rec}_i[F] \quad \text{s.t. } \text{cov}_j[F] \geq \frac{0.95}{K}, \forall j \in [K]. \tag{1}$$

Many of these metrics are **non-decomposable**, i.e., one cannot compute these metrics by simply calculating the average of scores on individual examples. Optimizing for these metrics can be regarded as instances of cost-sensitive learning (CSL). More specifically, *optimization problems of the form which can be written as a linear combination of $G_{i,j}$ and $C_{ij}[F]$ will be our focus in this work* where $\mathbf{G}$ is a $K \times K$ matrix.

$$\max_F \sum_{i,j \in [K]} G_{ij} C_{ij}[F], \tag{2}$$

The entry $G_{ij}$ represents the reward associated with predicting class $j$ when the true class is $i$. The matrix $\mathbf{G}$ is called a gain matrix [20]. Some more complex non-decomposable objectives for classification can be reduced to CSL [22, 32, 20]. For instance, the aforementioned two complex objectives can be reduced to CSL using continuous relaxation or a Lagrange multiplier as bellow. Let $\Delta_{K-1} \subset \mathbb{R}^K$ be the $K-1$-dimensional probability simplex. Then, maximizing the minimum recall is equivalent to the saddle-point optimization problem:

$$\max_F \min_{\boldsymbol{\lambda} \in \Delta_{K-1}} \sum_{i \in [K]} \lambda_i \frac{C_{ii}[F]}{\pi_i}. \tag{3}$$

Thus, for a fixed $\boldsymbol{\lambda}$, the corresponding gain matrix is given as a diagonal matrix $\text{diag}(G_1, \ldots, G_K)$ with $G_i = \lambda_i/\pi_i$ for $1 \leq i \leq K$. Similarly, using Lagrange multipliers $\boldsymbol{\lambda} \in \mathbb{R}_{\geq 0}^K$, Eq. (1) is rewritten as a max-min optimization problem [20, Sec. 2]:

$$\max_F \min_{\boldsymbol{\lambda} \in \mathbb{R}_{\geq 0}^K} \frac{1}{K} \sum_{i \in [K]} C_{ii}[F]/\pi_i + \sum_{j \in [K]} \lambda_j \left( \sum_{i \in [K]} C_{ij}[F] - 0.95/K \right). \tag{4}$$

In this case, the corresponding gain matrix $\mathbf{G}$ is given as $G_{ij} = \frac{\delta_{ij}}{K\pi_i} + \lambda_j$, where $\delta_{ij}$ is the Kronecker's delta. One can solve these max-min problems by alternatingly updating $\boldsymbol{\lambda}$ (using exponented gradient or projected gradient descent) and optimizing the cost-sensitive objectives [20].

## 2.2 Loss Functions for Non-Decomposable Objectives

The cross entropy loss function is appropriate for optimizing accuracy for deep neural networks, however, learning with CE can suffer low performance for cost-sensitive objectives [20]. Following [20], we introduce calibrated loss functions for given gain matrix $\mathbf{G}$. We let $p_m : \mathcal{X} \rightarrow \Delta_{K-1} \subset \mathbb{R}^K$ be a prediction function of a model, where $\Delta_{K-1}$ is the $K-1$-dimensional probability simplex. For a gain matrix $\mathbf{G}$, the corresponding loss function is given as a combination of logit adjustment [17] and loss re-weighting [24]. We decompose the gain matrix $\mathbf{G}$ as $\mathbf{G} = \mathbf{MD}$, where $\mathbf{D} = \text{diag}(G_{11}, \ldots, G_{KK})$ be a diagonal matrix, with $D_{ii} > 0, \forall i \in [K]$ and $\mathbf{M} \in \mathbb{R}^{K \times K}$. For $y \in [K]$ and model prediction $p_m(x)$, the hybrid loss is defined as follows:

$$\ell^{\text{hyb}}(y, p_m(x)) = -\sum_{i \in [K]} M_{yi} \log \left( \frac{(p_m(x))_i / D_{ii}}{\sum_{j \in [K]} (p_m(x))_j / D_{jj}} \right). \tag{5}$$

To make the dependence of $\mathbf{G}$ explicit, we also denote $\ell^{\text{hyb}}(y, p_m(x))$ as $\ell^{\text{hyb}}(y, p_m(x); \mathbf{G})$. The average loss on training sample $S \subset \mathcal{X}$ is defined as $\mathcal{L}^{\text{hyb}}(\mathcal{X}) = \frac{1}{|S|} \sum_{x \in S} \ell^{\text{hyb}}(u, p_m(x))$.

Narasimhan and Menon [20] proved that the hybrid loss is calibrated, that is learning with $\ell^{\text{hyb}}$ gives the Bayes optimal classifier for $\mathbf{G}$ (c.f., [20, Proposition 4], of which we provide a formal statement in Sec. N.1). If $\mathbf{G}$ is a diagonal matrix (i.e., $\mathbf{M} = \mathbf{1}_K$), the hybrid loss is called the logit adjusted (LA) loss and $\ell^{\text{hyb}}(y, p_m(x))$ is denoted by $\ell^{\text{LA}}(y, p_m(x))$.

## 2.3 Consistency Regularizer for Semi-Supervised Learning

Modern self-training methods not only leverage pseudo labels, but also forces consistent predictions of a classifier on augmented examples or neighbor examples [35, 18, 36, 31]. More formally, a classifier $F$ is trained so that the consistent regularizer $R(F)$ is small while a supervised loss or a loss between pseudo labeler are minimized [35, 31]. Here the consistency regularizer $R(F)$ is defined as

$$\mathbf{E}_x \left[ \mathbb{1}(F(x) \neq F(x'), \exists x' \text{ s.t. } x' \text{ is a neighbor of an augmentation of } x) \right].$$

In existing works, consistency regularizers are considered for optimization of accuracy. In the subsequent sections, we consider consistency regularizers for cost-sensitive objectives.

# 3 Cost-Sensitive Self-Training for Non-Decomposable Metrics

## 3.1 CSL and Weighted Error

In the case of accuracy or 0-1-error, a self-training based SSL algorithm using a consistency regularizer achieves the state-of-the-art performance across a variety of datasets [31] and its effectiveness has been proved theoretically [35]. This section provides theoretical analysis of a self-training based SSL algorithm for non-decomposable objectives by generalizing [35]. More precisely, the main result of this section (Theorem 5) states that an SSL method using consistency regularizer improves a given pseudo labeler for non-decomposable objectives. We provide all the omitted proofs in Appendix for theoretical results in the paper.

In Sec. 2, we considered non-decomposable metrics and their reduction to cost-sensitive learning objectives defined by Eq. (2) using a gain matrix. In this section, we consider an equivalent objective using the notion of weighted error. For weight matrix $w = (w_{ij})_{1 \leq i, j \leq K}$ and a classifier $F : \mathcal{X} \to [K]$, a weighted error is defined as follows:

$$\text{Err}_w(F) = \sum_{i,j \in [K]} w_{ij} \mathbf{E}_{x \sim P_i} \left[ \mathbb{1}(F(x) \neq j) \right],$$

where, $P_i(x)$ denotes the class conditional distribution $\mathbf{P}(x \mid y = i)$. If $w = \text{diag}(1/K, \ldots, 1/K)$, then this coincides with the usual balanced error [17]. Since $C_{ij}[F] = \mathbf{E}_{(x,y) \sim D} \left[ \mathbb{1}(y = i, F(x) = j) \right] = \mathbf{P}(y = i) - \mathbf{P}(y = i) \mathbf{E}_{x \sim P_i} \left[ \mathbb{1}(F(x) \neq j) \right]$, we can write:

$$G_{ij} C_{ij}[F] = G_{ij}(\mathbf{P}(y = i) - \mathbf{P}(y = i) \mathbf{E}_{x \sim P_i} \left[ \mathbb{1}(F(x) \neq j) \right]) = G_{ij}(\pi_i - \pi_i \mathbf{E}_{x \sim P_i} \left[ \mathbb{1}(F(x) \neq j) \right])$$

Here $\pi_i$ is the class prior $\mathbf{P}(y = i)$ for $1 \leq i \leq K$ as before. Hence CSL objective (2) i.e. $\max_F \sum_{i,j} G_{ij} C_{ij}[F]$ is equivalent to minimizing the negative term above i.e. $G_{ij} \pi_i \mathbf{E}_{x \sim P_i} \left[ \mathbb{1}(F(x) \neq j) \right]$ which is same as $\text{Err}_w(F)$ with $w_{ij} = G_{ij} \pi_i$ for $1 \leq i, j \leq K$. Hence, the *notion of weighted error is equivalent to CSL*, which we will also use later for deriving loss functions. We further note that if we add a matrix with the same columns ($c\mathbb{1} \geq 0$) to the gain matrix **G**, still the maximizers of CSL (2) are the same as the original problem. Hence, without loss of generality, we assume $w_{ij} \geq 0$. We assume $w \neq \mathbf{0}$, i.e., $|w|_1 > 0$ for avoiding degenerate solutions.

In the previous work [35], it is assumed that there exists a ground truth classifier $F^\star : \mathcal{X} \to [K]$ and the supports of distributions $\{P_i\}_{1 \leq i \leq K}$ are disjoint. However, if supports of distributions $\{P_i\}_{1 \leq i \leq K}$ are disjoint, a solution of the minimization problem $\min_F \text{Err}_w(F)$ is independent of $w$ in some cases. More precisely, if $w = \text{diag}(w_1, \ldots, w_K)$ i.e. a diagonal matrix and $w_i > 0, \forall i$, then the optimal classifier is given as $x \mapsto \text{argmax}_{k \in [K]} w_k P_k(x)$ (this follows from [20, Proposition 1]). If supports are disjoint, then the optimal classifier is the same as $x \mapsto \text{argmax}_{k \in [K]} P_k(x)$, which coincides with the ground truth classifier. Therefore, we do not assume the supports of $P_i$ are disjoint nor a ground truth classifier exists unlike [35]. See Fig. 2 for an intuitive explanation.

## 3.2 Weighted Consistency Regularizer

For improving accuracy, the consistency in prediction is equally important across the distributions $P_i$ for $1 \leq i \leq K$ [31, 35]. However, for our case of weighted error, if the entries of the $i_0^{th}$ row of the weight matrix $w$ are larger than the other entries for some $i_0$, then the consistency of model predictions on examples drawn from the distribution $P_{i_0}$ are more important than those on the other

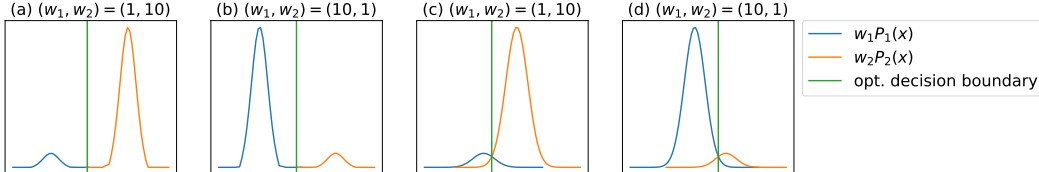

Figure 2: Using a simple example, we explain a difference in theoretical assumptions compared to [35] that assumes $\{P_i\}_{1\leq i \leq K}$ have disjoint supports. Here, we consider the case when $K = 2$, $w = \mathrm{diag}(w_1, w_2)$, and $P_1$ and $P_2$ are distributions on $\mathcal{X} \subset \mathbb{R}$. (a), (b): In a perfect setting where two distributions $P_1$ and $P_2$ have disjoint supports, the Bayes optimal classifier for the CSL is identical to the ground truth classifier ($x \mapsto \mathrm{argmax}_i P_i(x)$) for any choices of weights $(w_1, w_2)$. (c), (d): In a more generalized setting, the Bayes optimal classifier $x \mapsto \mathrm{argmax}_i w_i P_i(x)$ depends on the choice of weights (i.e., gain matrix). The optimal decision boundary (the green line) for the CSL moves to the right as $w_1/w_2$ increases.

examples. In this case, we require more restrictive consistency regularizer for distribution for $P_{i_0}$. Thus, we require a weighted (or cost-sensitive) consistency regularizer, which we define below.

We assume that the instance space $\mathcal{X}$ is a normed vector space with norm $|\cdot|$ and $\mathcal{T}$ is a set of augmentations, i.e., each $T \in \mathcal{T}$ is a map from $\mathcal{X}$ to itself. For a fixed $r > 0$, we define $\mathcal{B}(x)$ by $\{x' \in \mathcal{X} : \exists T \in \mathcal{T} \text{ s.t. } |x' - T(x)| \leq r\}$. For each $i \in [K]$, we define conditional consistency regularizer by $R_{\mathcal{B},i}(F) = \mathbf{E}_{x \sim P_i}\left[\mathbb{1}\left(\exists x' \in \mathcal{B}(x) \text{ s.t. } F(x) \neq F(x')\right)\right]$. Then, we define the weighted consistency regularizer by $R_{\mathcal{B},w}(F) = \sum_{i,j \in [K]} w_{ij} R_{\mathcal{B},i}(F)$. If the prediction of classifier $F$ is robust to augmentation $T \in \mathcal{T}$ and small noise, then $R_{\mathcal{B},w}(F)$ is small. For $\beta > 0$, we consider the following optimization objective for finding a classifier $F$:

$$\min_F \mathrm{Err}_w(F) \quad \text{subject to } R_{\mathcal{B},w}(F) \leq \beta. \tag{6}$$

A solution of the problem (6) is denoted by $F^*$. We let $F_{\mathrm{pl}} : \mathcal{X} \to [K]$ a pseudo labeler (a classifier) with reasonable performance (we elaborate on this in Section 3.4). The following mathematically informal assumption below is required to interpret our main theorem.

**ASSUMPTION 1.** We assume both $\beta$ and $\mathrm{Err}_w(F^*)$ are sufficiently small so that they are negligible compared to $\mathrm{Err}_w(F_{\mathrm{pl}})$.

Assumption 1 assumes existence of an optimal classifier $F^*$ that minimizes the error $\mathrm{Err}_w(F)$ (i.e. Bayes Optimal) among the class of classifiers which are robust to data augmentation (i.e. low weighted consistency $R_{\mathcal{B},w}(F)$). As we operate in case of overparameterized neural networks such a classifier $F^*$ is bound to exist, but is unknown in our problem setup. In the case of the balanced error, the validity of this assumption is justified by the fact that the existing work [31] using consistency regularizer on data augmentation obtains classifier $F$, that achieves high accuracy (i.e., low balanced errors) for balanced datasets. Also in Appendix C.1, we provide an example that supports the validity of the assumption in the case of Gaussian mixtures and diagonal weight matrices.

### 3.3 Expansion Property

For $x \in \mathcal{X}$, we define the neighborhood $\mathcal{N}(x)$ of $x$ by $\{x' \in \mathcal{X} : \mathcal{B}(x) \cap \mathcal{B}(x') \neq \emptyset\}$. For a subset $S \subseteq \mathcal{X}$, neighborhood of $S$ is defined as $\mathcal{N}(S) = \cup_{x \in S} \mathcal{N}(x)$. Similarly to [35], we consider the following property on distributions.

**DEFINITION 2.** Let $c : (0, 1] \to [1, \infty)$ be a non-increasing function. For a distribution $Q$ on $\mathcal{X}$ we say $Q$ has $c$-expansion property if $Q(\mathcal{N}(S)) \geq c(Q(S))Q(S)$ for any measurable $S \subseteq \mathcal{X}$.

The $c$-expansion property implies that if $Q(S)$ decreases, then the "expansion factor" $Q(\mathcal{N}(S))/Q(S)$ increases. This is a natural condition, because it roughly requires that if $Q(S)$ is small, then $Q(\mathcal{N}(S))$ is large compared to $Q(S)$. For intuition let us consider a ball of radius $l$ depicting $S \subset \mathbb{R}^d$ with volume $Q(S)$ and its neighborhood $\mathcal{N}(S)$ expands to a ball with radius $l + 1$. The expansion factor here would be $((l + 1)/l)^d$, hence as $l$ (i.e. $Q(S)$) increases $(1 + 1/l)^d$ (i.e. $Q(\mathcal{N}(S))/Q(S)$) decreases. Hence, it's natural to expect $c$ to be a non-decreasing function. The $c$-expansion property (on each $P_i$) considered here is equivalent to the $(a, \widetilde{c})$-expansion property, which is shown to be

realistic for vision and used for theoretical analysis of self-training in [35], where $a \in (0, 1)$ and $\widetilde{c} > 1$ (see Sec. N.2 in Appendix). In addition, we also show that it is also satisfied for mixtures of Gaussians and mixtures of manifolds (see Example 1 in Appendix for more details). Thus, the $c$-expansion property is a general property satisfied for a wide class of distributions.

**ASSUMPTION 3.** For weighted probability measure $\mathcal{P}_w$ on $\mathcal{X}$ by $\mathcal{P}_w(U) = \frac{\sum_{i,j\in[K]} w_{ij} P_i(U)}{\sum_{i,j\in[K]} w_{ij}}$ for $U \subseteq \mathcal{X}$. We assume $\mathcal{P}_w$ satisfies $c$-expansion for a non-increasing function $c : (0, 1] \rightarrow [1, \infty)$.

### 3.4 Cost-Sensitive Self-Training with Weighted Consistency Regularizer

In this section we first introduce the assumptions on the pseudo labeler $F_{\mathrm{pl}}$ and then introduce the theoretical Cost-Sensitive Self-Training (CSST) objective. $F_{\mathrm{pl}}$ can be any classifier satisfying the following assumption, however, typically it is a classifier trained on a labeled dataset.

**ASSUMPTION 4.** We assume that $\mathrm{Err}_w(F_{\mathrm{pl}}) + \mathrm{Err}_w(F^*) \leq |w|_1$. Let $\gamma = c(p_w)$, where $p_w = \frac{\mathrm{Err}_w(F_{\mathrm{pl}}) + \mathrm{Err}_w(F^*)}{|w|_1}$. We also assume $\gamma > 3$.

Since $c$ is non-increasing, $\gamma$ (as a function of $\mathrm{Err}_w(F_{\mathrm{pl}})$) is a non-increasing function of $\mathrm{Err}_w(F_{\mathrm{pl}})$ (and $\mathrm{Err}_w(F^*)$). Therefore, the assumption $\gamma > 3$ roughly requires that $\mathrm{Err}_w(F_{\mathrm{pl}})$ is "small". We provide concrete conditions for $\mathrm{Err}_w(F_{\mathrm{pl}})$ that satisfy $\gamma > 3$ in the case of mixture of isotropic $d$-dimenional Gaussians for a region $\mathcal{B}(x)$ defined by $r$ in Appendix (Example 2). In the example, we show that the condition $\gamma > 3$ is satisfied if $\mathrm{Err}_w(F_{\mathrm{pl}}) < 0.17$ in the case when $r = 1/(2\sqrt{d})$ and satisfied if $\mathrm{Err}_w(F_{\mathrm{pl}}) < 0.33$ in the case when $r = 3/(2\sqrt{d})$, where $\mathcal{X} \subseteq \mathbb{R}^d$. Since we assume $\mathrm{Err}_w(F^*)$ is negligible compared to $\mathrm{Err}_w(F_{\mathrm{pl}})$ (Assumption 1), the former condition in Assumption 4 is approximately equivalent to $\mathrm{Err}_w(F_{\mathrm{pl}}) \leq |w|_1$ which is satisfied by the definition of $\mathrm{Err}_w$.

We define $L_{0\text{-}1}^{(i)}(F, F') = \mathbf{E}_{x \sim P_i} [\mathbb{1}(F(x) \neq F'(x))]$. Then, we consider the following objective:

$$\min_F \mathcal{L}_w(F), \quad \text{where } \mathcal{L}_w(F) = \frac{\gamma+1}{\gamma-1} L_w(F, F_{\mathrm{pl}}) + \frac{2\gamma}{\gamma-1} R_{\mathcal{B},w}(F). \tag{7}$$

Here $L_w(F, F_{\mathrm{pl}})$ is defined as $\sum_{i,j\in[K]} w_{ij} L_{0\text{-}1}^{(i)}(F, F_{\mathrm{pl}})$. The above objective corresponds to cost-sensitive self-training (with $F_{\mathrm{pl}}$) objective with weighted consistency regularization. We provide following theorem which relates the weighted error of classifier $\hat{F}$ learnt using the above objective to the weighted error of the pseudo labeler ($F_{\mathrm{pl}}$).

**THEOREM 5.** *Any learnt classifier $\widehat{F}$ using the loss function $\mathcal{L}_w$ (i.e., $\arg\min_F \mathcal{L}_w(F)$) satisfies:*

$$\mathrm{Err}_w(\widehat{F}) \leq \frac{2}{\gamma-1}\mathrm{Err}_w(F_{\mathrm{pl}}) + \frac{\gamma+1}{\gamma-1}\mathrm{Err}_w(F^*) + \frac{4\gamma}{\gamma-1}R_{\mathcal{B},w}(F^*).$$

**REMARK.** Since both $\mathrm{Err}_w(F^*)$ and $R_{\mathcal{B},w}(F^*) \leq \beta$ are negligible compared to $\mathrm{Err}_w(F_{\mathrm{pl}})$ and $\gamma > 3$, Theorem 5 asserts that $\hat{F}$ learnt by minimizing semi-supervised loss $L_w(F, F_{\mathrm{pl}})$ with the consistency regularizer $R_{\mathcal{B},w}(F)$ can achieve superior performance than the pseudo labeler $F_{\mathrm{pl}}$ in terms of the weighted error $\mathrm{Err}_w$. The above theorem is a generalization of [35, Theorem 4.3], which provided a similar result for balanced 0-1-error in the case of distributions with disjoint supports. In Appendix Sec. D, following [35, 34], we also provide a generalization bound for $\mathrm{Err}_w(F)$ using all-layer margin [34] in the case when classifiers are neural networks.

## 4 CSST in Practice

In the previous section, we proved that by using self-training (CSST), we can achieve a superior classifier $\hat{F}$ in comparison to pseudo labeler $F_{\mathrm{pl}}$ through weighted consistency regularization. As we have established the equivalence of the weighted error $Err_w$ to the CSL objective expressed in terms of $\mathbf{G}$ (Sec. 3.1) , we can theoretically optimize a given non-decomposable metric expressed by $\mathbf{G}$ better using CSST, utilizing the additional unlabeled data via self-training and weighted consistency regularization. We now show how CSST can be used in practice for optimizing non-decomposable metrics in the case of neural networks.

The practical self-training methods utilizing consistency regularization (e.g., FixMatch [31], etc.) for semi-supervised learning have supervised loss $\mathcal{L}_s$ for labeled and consistency regularization loss for

unlabeled samples (i.e., $\mathcal{L}_u$) with a thresholding mechanism to select unlabeled samples. The final loss for training the network is $\mathcal{L}_s + \lambda_u \mathcal{L}_u$, where $\lambda_u$ is the hyperparameter. The supervised loss $\mathcal{L}_s$ can be modified conveniently based on the desired non-decomposable metric by using suitable $\mathbf{G}$ (Sec. 2.1). We will now introduce the novel weighted consistency loss and its corresponding thresholding mechanism for unlabeled data in CSST, used for optimizing desired non-decomposable metric.

**Weighted Consistency Regularization.** As the idea of consistency regularization is to enforce consistency between model prediction on different augmentations of input, this is usually achieved by minimizing some kind of divergence $\mathcal{D}$. A lot of recent works [18, 31, 36] in semi-supervised learning use $\mathcal{D}_{\mathrm{KL}}$ to enforce consistency between the model's prediction on unlabeled data and its augmentations, $p_m(x)$ and $p_m(\mathcal{A}(x))$. Here $\mathcal{A}$ usually denotes a form of strong augmentation. Across these works, the distribution of confidence of the model's prediction is either sharpened or used to get a hard pseudo label to obtain $\hat{p}_m(x)$. As we aim to optimized the cost-sensitive learning objective, we aim to match the distribution of normalized distribution (i.e. $\mathrm{norm}(\mathbf{G}^{\mathbf{T}}\hat{p}_m(x)) = \mathbf{G}^{\mathbf{T}}\hat{p}_m(x)/\sum_i (\mathbf{G}^{\mathbf{T}}\hat{p}_m(x))_i$) (Proposition 2 [20] also in Prop. 7 ) with the $p_m(\mathcal{A}(x))$ by minimizing the KL-Divergence between these. We now propose to use the following weighted consistency regularizer loss function for optimizing the same:

$$\ell_u^{\mathrm{wt}}(\hat{p}_m(x), p_m(\mathcal{A}(x)), \mathbf{G}) = -\sum_{i=1}^{K}(\mathbf{G}^{\mathbf{T}}\hat{p}_m(x))_i \log(p_m(\mathcal{A}(x))_i) \tag{8}$$

**PROPOSITION 6.** *The minimizer of* $\mathcal{L}_u^{wt} = \frac{1}{|B_u|}\sum_{x \in B_u} \ell_u^{wt}(\hat{p}_m(x), p_m(\mathcal{A}(x)), \mathbf{G})$ *leads to minimization of KL Divergence i.e.* $\mathcal{D}_{KL}(\mathrm{norm}(\mathbf{G}^{\mathbf{T}}\hat{p}_m(x))||p_m(\mathcal{A}(x))) \ \forall x \in \mathcal{X}$ .

As the above loss is similar in nature to the cost sensitive losses introduced by Narasimhan and Menon [20] (Sec. 2.1) we can use the logit-adjusted variants (i.e. $\ell^{\mathrm{LA}}$ and $\ell^{\mathrm{hyb}}$ based on type of $G$) of these in our final loss formulations ($\mathcal{L}_u^{wt}$) for training overparameterized deep networks. We further show in Appendix Sec. B that the above loss $\ell_u^{\mathrm{wt}}$ approximately minimizes the theoretical weighted consistency regularization term $R_{\mathcal{B},w}(F)$ defined in Sec. 3.4.

**Threshold Mechanism for CSST.** In the usual semi-supervised learning formulation [31] we use the confidence threshold ($\max_i(p_m(x)_i) > \tau$) as the function to select samples for which consistency regularization term is non-zero. We find that this leads to sub-optimal results in particularly the case of non-diagonal $\mathbf{G}$ as only a few samples cross the threshold (Fig. 4). As in the case of cost-sensitive loss formulation the samples may not achieve the high confidence to cross the threshold of consistency regularization. This is also theoretically justified by the following proposition:

**PROPOSITION 7** ([20] Proposition 2). *Let $p_m^{opt}(x)$ be the optimal softmax model function obtained by optimizing the cost-sensitive objective in Eq. (2) by averaging weighted loss function $\ell^{wt}(y, p_m(x)) = -\sum_{i=1}^{K} G_{y,i} \log \frac{(p_m(x)_i)}{\sum_j p_m(x)_j}$. Then optimal $p_m^{opt}(x)$ is: $p_m^{opt}(x) = \frac{G_{y,i}}{\sum_j G_{y,j}} = \mathrm{norm}(\mathbf{G}^T\mathbf{y})\forall(x,y)$.*

Here $\mathbf{y}$ is the one-hot representation vector for a label $y$. This proposition demonstrates that for a particular sample the high confidence $p_m(x)$ may not be optimal based on $\mathbf{G}$. We now propose our novel way of thresholding samples for which consistency regularization is applied in CSST. Our thresholding method takes into account the objective of optimizing the non-decomposable metric by taking $\mathbf{G}$ into account. We propose to use the threshold on KL-Divergence of the softmax of the sample $p_m(x)$ with the optimal softmax (i.e. $\mathrm{norm}(\mathbf{G}^T\hat{p}_m(x))$) for a given $\mathbf{G}$ corresponding to the pseudo label (or sharpened) $\hat{p}_m(x)$, using which we modify the consistency regularization loss term:

$$\mathcal{L}_u^{wt}(B_u) = \frac{1}{|B_u|}\sum_{x \in B_u} \mathbb{1}_{(\mathcal{D}_{KL}(\mathrm{norm}(\mathbf{G}^T\hat{p}_m(x)) \ || \ p_m(x)) \leq \tau)}\ell_u^{\mathrm{wt}}(\hat{p}_m(x), p_m(\mathcal{A}(x)), \mathbf{G}) \tag{9}$$

We name this proposed combination of KL-Thresholding and weighted consistency regularization as CSST in our experimental results. We find that for non-diagonal gain matrix $\mathbf{G}$ the proposed thresholding plays a major role in improving performance over supervised learning. This is demonstrated by comparison of CSST and CSST w/o KL-Thresholding (without proposed thresholding mechanism) in Fig. 4 and Tab. 3. We will now empirically incorporate CSST by introducing consistency based losses and thresholding mechanism for unlabeled data, into the popular semi-supervised methods of FixMatch [31] and Unsupervised Data Augmentation for Consistency Training (UDA) [36]. The exact expression for the weighted consistency losses utilized for UDA and FixMatch have been provided in the Appendix Sec. H.2 and H.3 .

Table 2: Results of maximizing the worst-case recall over *all classes* (col 2–3) and over just the head and tail classes (col 4–7).

| Method | CIFAR10-LT ($\rho = 100$) | | CIFAR100-LT ($\rho = 10$) | | Imagenet100-LT ($\rho = 10$) | |
|---|---|---|---|---|---|---|
| | Avg. Rec | Min. Rec | Avg. Rec | Min. HT Rec | Avg. Rec | Min. HT Rec |
| ERM | 0.52 | 0.26 | 0.36 | 0.14 | 0.40 | 0.30 |
| LA | 0.51 | 0.38 | 0.36 | 0.35 | 0.48 | 0.47 |
| CSL | 0.64 | 0.57 | 0.43 | 0.43 | 0.52 | 0.52 |
| Vanilla (FixMatch) | 0.78 | 0.48 | 0.63 | 0.36 | 0.58 | 0.49 |
| CSST (FixMatch) | 0.76 | 0.72 | 0.63 | 0.61 | 0.64 | 0.63 |

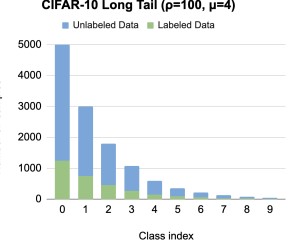

Figure 3: CIFAR-10 Long tail distribution $\rho = 100, \mu = 4$.

## 5   Experiments

We demonstrate that the proposed CSST framework shows significant gains in performance on both vision and NLP tasks on imbalanced datasets, with an imbalance ratio defined on the training set as $\rho = \frac{\max_i P(y=i)}{\min_i P(y=i)}$. We assume the labeled and unlabeled samples come from a similar data distribution and the unlabeled samples are much more abundant ($\mu$ times) the labeled. The frequency of samples follows an exponentially decaying long-tailed distributed as seen in Fig. 3, which closely imitates the distribution of real-world data [33, 10]. For CIFAR-10 [11], IMDb [16] and DBpedia-14 [14], we use $\rho = 100$ and $\rho = 10$ for CIFAR-100 [11] and ImageNet-100 [27] [2] datasets. We compare our method against supervised methods of ERM, Logit Adjustment (LA) [17] and Cost Sensitive Learning (CSL) [21] trained on the same number of labeled samples as used by semi-supervised learning methods, along with vanilla semi-supervised methods of FixMatch (for vision) and UDA (for NLP tasks). We use WideResNets(WRN) [39], specifically WRN-28-2 and WRN-28-8 for CIFAR-10 and CIFAR-100 respectively. For ImageNet, we use a ResNet-50 [7] network for our experiments and finetuned DistilBERT(base uncased) [29] for IMDb and DBpedia-14 datasets. We divide the balanced held-out set for each dataset equally into validation and test sets. A detailed list of hyper-parameters and additional experiments can be found in the Appendix Tab. 4 and Sec. O respectively.

**Maximizing Worst-Case Recall.** For CIFAR-10, IMDb, and DBpedia-14 datasets, we maximize the minimum recall among all classes (Eq. (3)). Given the low number of samples per class for datasets with larger number of classes like CIFAR-100 and ImageNet-100, we pick objective (10). We define the head classes ($\mathcal{H}$) and tail classes ($\mathcal{T}$) as the first 90 classes and last 10 classes respectively. The Min. HT recall objective can be mathematically formulated as:

$$\max_F \min_{(\lambda_{\mathcal{H}}, \lambda_{\mathcal{T}}) \in \Delta_1} \frac{\lambda_{\mathcal{H}}}{|\mathcal{H}|} \sum_{i \in \mathcal{H}} \frac{C_{ii}[F]}{\pi_i} + \frac{\lambda_{\mathcal{T}}}{|\mathcal{T}|} \sum_{i \in \mathcal{T}} \frac{C_{ii}[F]}{\pi_i}. \tag{10}$$

The corresponding gain matrix $\mathbf{G}$ is diag($\frac{\lambda_{\mathcal{H}}}{\pi_1|\mathcal{H}|}, \frac{\lambda_{\mathcal{H}}}{\pi_2|\mathcal{H}|}, \ldots, \frac{\lambda_{\mathcal{T}}}{\pi_{K-1}|\mathcal{T}|}, \frac{\lambda_{\mathcal{T}}}{\pi_K|\mathcal{T}|}$). Since $\mathbf{G}$ is diagonal here, we use CSST(FixMatch) loss function Eq. (9) with the corresponding $\ell_u^{\text{wt}}$ being substituted by $\ell_u^{\text{LA}}$ as define in Sec. 2.1. Also for labeled samples we use $\mathcal{L}_s^{\text{LA}}$ as $\mathbf{G}$ is diagonal, we then combine the loss and train network using SGD. Each few steps of SGD, were followed by an update on the $\boldsymbol{\lambda}$ and $\mathbf{G}$ based on the uniform validation set (See Alg. 1 in Appendix). We find that CSST(FixMatch) significantly outperforms the other baselines in terms of the Min. recall and Min. Head-Tail recall for all datasets, the metrics which we aimed to optimize (Tab. 2), which shows effectiveness of CSST framework. Despite optimizing worst-case recall we find that our framework is still able to maintain reasonable average (Avg.) recall in comparison to baseline vanilla(FixMatch), which demonstrates it's practical applicability. We find that optimizing Min. recall across NLP tasks of classification on long-tailed data by plugging UDA into CSST(UDA) framework shows similar improvement in performance (Tab. 4). This establishes the generality of our framework to even self-training methods across domain of NLP as well. As the $\mathbf{G}$ is a diagonal matrix for this objective, the proposed KL-Based Thresholding here is equivalent to the confidence based threshold of FixMatch in this case. Despite the equivalence of thresholding mechanism, we see significant gains in min-recall (Tab. 2) just using the regularization term. We discuss their equivalance in Appendix Sec. I.

**Maximizing Mean Recall Under Coverage Constraints.** Maximizing mean recall under coverage constraints objective seeks to result in a model with good average recall, yet at the same time

---

[2]https://www.kaggle.com/datasets/ambityga/imagenet100

Table 3: Results of maximizing the mean recall subject to coverage constraint *all classes* (col 2–3) and over the head and tail classes (col 4–7). Proposed CSST(FixMatch) approach compares favorably to ERM,LA,CSL vanilla(FixMatch) and CSST(FixMatch) w/o KL-Thresh.. It is the best at both maximizing mean recall and coming close to satisfying the coverage constraint.

| Method | CIFAR10-LT Per-class Coverage ($\rho = 100$, tgt : 0.1) | | CIFAR100-LT Head-Tail Coverage ($\rho = 10$, tgt : 0.01) | | ImageNet100-LT Head-Tail Coverage ($\rho = 10$, tgt : 0.01) | |
|---|---|---|---|---|---|---|
| | Avg. Rec | Min. Cov | Avg. Rec | Min. HT Cov | Avg. Rec | Min. HT Cov |
| ERM | 0.52 | 0.034 | 0.36 | 0.004 | 0.40 | 0.006 |
| LA | 0.51 | 0.039 | 0.36 | 0.009 | 0.48 | 0.009 |
| CSL | 0.60 | 0.090 | 0.45 | 0.010 | 0.48 | 0.010 |
| Vanilla (FixMatch) | 0.78 | 0.055 | 0.63 | 0.004 | 0.58 | 0.007 |
| CSST(FixMatch) w/o KL-Thresh. | 0.67 | 0.093 | 0.47 | 0.010 | 0.26 | 0.010 |
| CSST(FixMatch) | 0.80 | 0.092 | 0.63 | 0.010 | 0.58 | 0.010 |

Table 4: Results of maximizing the min recall over *all classes* for classification on NLP datasets. Proposed CSST(UDA) approach outperforms ERM and vanilla(UDA) baselines.

| Method | IMDb ($\rho = 10$) | | IMDb ($\rho = 100$) | | DBpedia-14 ($\rho = 100$) | |
|---|---|---|---|---|---|---|
| | Avg Rec | Min Rec | Avg Rec | Min Rec | Avg Rec | Min Rec |
| ERM | 0.79 | 0.61 | 0.50 | 0.00 | 0.95 | 0.58 |
| vanilla(UDA) | 0.82 | 0.66 | 0.50 | 0.00 | 0.96 | 0.65 |
| CSST(UDA) | 0.89 | 0.88 | 0.77 | 0.75 | 0.99 | 0.97 |

constraints the proportion of predictions for each class to be uniform across classes. The ideal target coverage under a balanced evaluation set(or such circumstances) is given as $\text{cov}_i[F] = \frac{1}{K}, \forall i \in [K]$. Along similar lines to objective (10) we modify the objective (4) to maximize the average recall subject to the both the average head and tail class coverage (HT Coverage) being above a given threshold of $\frac{0.95}{K}$. The **G** is non-diagonal here and $G_{ij} = (\mathbb{1}_{j \in \mathcal{H}} \frac{\lambda_{\mathcal{H}}}{|\mathcal{H}|} + \mathbb{1}_{j \in \mathcal{T}} \frac{\lambda_{\mathcal{T}}}{|\mathcal{T}|}) + \frac{\delta_{ij}}{K \pi_i}$.

$$\max_F \min_{(\lambda_{\mathcal{H}}, \lambda_{\mathcal{T}}) \in \mathbb{R}^2_{\geq 0}} \sum_{i \in [K]} \frac{C_{ii}[F]}{K \pi_i} + \lambda_{\mathcal{H}} \left( \sum_{i \in [K], j \in \mathcal{H}} \frac{C_{ij}[F]}{|\mathcal{H}|} - \frac{0.95}{K} \right) + \lambda_{\mathcal{T}} \left( \sum_{i \in [K], j \in \mathcal{T}} \frac{C_{ij}[F]}{|\mathcal{T}|} - \frac{0.95}{K} \right).$$
(11)

As these objectives corresponds to a non-diagonal **G** as shown in Eq. (4) in Sec. 2.1. Hence, for introducing CSST into FixMatch we replace first supervised loss $\mathcal{L}_s$ with $\mathcal{L}_s^{\text{hyb}}$. For the unlabeled data we introduce $\ell^{\text{hyb}}$ in $\mathcal{L}_u^{\text{wt}}$ (Eq. (9)). Hence, the final objective $\mathcal{L}$ is defined as, $\mathcal{L} = \mathcal{L}_s^{\text{hyb}} + \lambda_u \mathcal{L}_u^{\text{wt}}$. We update the parameters of the cost-sensitive loss (**G** and $\lambda$) periodically after few of SGD on the model parameters (Alg. 2 in Appendix). In this case our proposed thresholding mechanism in CSST(FixMatch) introduced in Sec. 4, leads to effective utilization of unlabled data resulting in improved performance over the naive CSST(FixMatch) without (w/o) KL-Thresholding (Tab. 3). In these experiments, the mean recall of our proposed approach either improves or stays same to the vanilla(FixMatch) implementation but only ours is the one that comes close to satisfying the coverage constraint. We further note that among all the methods, the only methods that come close to satisfying the coverage constraints are the ones with $\ell^{\text{hyb}}$ included in them.

**Ablation on amount of unlabeled data.** Here, we ablated the total number of labeled sampled while keeping the number of unlabeled samples constant. We observe (in Fig. 5a) that as we increase the number of labeled samples the mean recall improved. This is because more labeled samples helps in better pseudo-labeling on the unlabeled samples and similarly as we decrease the number of labeled sample, the models errors on the pseudo-labels increase causing a reduction in mean recall. Hence, any addtional labeled data can be easily used to improve CSST performance.

**Ablation on $\tau$ threshold.** Fig. 5b shows that when the KL divergence threshold is too high, a large number of samples with very low degree of distribution match are used for generating the sharpened target (or pseudo-labels), this leads to worsening of mean recall as many targets are incorrect. We find that keeping a conservative of $\tau = 0.3$ works well across multiple experiments.

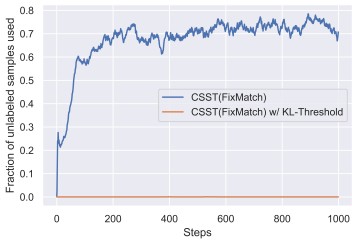 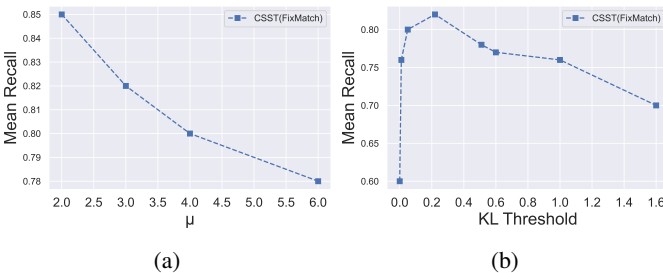

(a)                                            (b)

Figure 4: Fraction of unlabelled data used for maximizing average recall under coverage constraints for CIFAR-100 ($\rho = 10, \mu = 4$) (Sec. 5).

Figure 5: Maximizing average recall under coverage constraints for CIFAR-10 Long tail ($\rho = 100$) (Sec. 5). Fig. shows comparison of (a) increasing the ratio of unlabeled samples to labeled samples given fixed number of unlabeled samples (b) Ablation on KL diveregence based threshold for CSST(FixMatch)

## 6 Related Work

**Self-Training.** Self-training algorithms have been popularly used for the tasks of semi-supervised learning [1, 37, 31, 13] and unsupervised domain adaptation [28, 41]. In recent years several regularizers which enforce consistency in the neighborhood (either an adversarial perturbation [18] or augmentation [36]) of a given sample have further enhanced the applicability and performance of self-training methods, when used in conjunction. However, these works have focused mostly on improving the generic metric of accuracy, unlike the general non-decomposable metrics we consider.

**Cost-Sensitive Learning.** It refers to problem settings where the cost of error differs for a sample based on what class it belongs to. These settings are very important for critical real world applications like disease diagnosis, wherein mistakenly classifying a diseased person as healthy can be disastrous. There have been a plethora of techniques proposed for these which can be classified into: importance weighting [15, 38, 5] and adaptive margin [2, 38] based techniques. For overparameterized models Narasimhan et al. [22] show that loss weighting based techniques are ineffective and propose a logit-adjustment based cost-sensitive loss which we also use in our framework.

**Complex Metrics for Deep Learning.** There has been a prolonged effort on optimizing more complex metrics that take into account practical constraints [21, 26, 23]. However most work has focused on linear models leaving scope for works in context of deep neural networks. Sanyal et al. [30] train DNN using reweighting strategies for optimizing metrics, Huang et al. [8] use a reinforcement learning strategy to optimize complex metrics, and Kumar et al. [12] optimize complex AUC (Area Under Curve) metric for a deep neural network. However, all these works have primarily worked in supervised learning setup and are not designed to effectively make use of available unlabeled data.

## 7 Conclusion

In this work, we aim to optimize practical non-decomposable metrics readily used in machine learning through self-training with consistency regularization, a class of semi-supervised learning methods. We introduce a cost-sensitive self-training framework (CSST) that involves minimizing a cost-sensitive error on pseudo labels and consistency regularization. We show theoretically that we can obtain classifiers that can better optimize the desired non-decomposable metric than the original model used for obtaining pseudo labels, under similar data distribution assumptions as used for theoretical analysis of Self-training. We then apply CSST to practical and effective self-training method of FixMatch and UDA, incorporating a novel regularizer and thresholding mechanism based on a given non-decomposable objective. We find that CSST leads to a significant gain in performance of desired non-decomposable metric, in comparison to vanilla self-training-based baseline. Analyzing the CSST framework when the distribution of unlabeled data significantly differs from labeled data is a good direction to pursue for future work.

**Acknowledgements:** This work was supported in part by Fujitsu Research grant. Harsh Rangwani is supported by Prime Minister's Research Fellowship (PMRF).

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
