# Cost-Sensitive Self-Training for Optimizing Non-Decomposable Metrics

# Contents

Submitted to 36th Conference on Neural Information Processing Systems (NeurIPS 2022). Do not distribute.

## A    Limitations and Negative Societal Impacts

### A.1    Limitations of our Work

At this point we only consider optimizing objectives through CSST which can be written as a linear combination of entries of confusion matrix. Although there are important metrics like Recall, Coverage etc. which can be expressed as a linear form of confusion matrix. However, there do exist important metrics like Intersection over Union (IoU), Q-Mean etc. which we don't consider in the current work. We leave this as an open direction for further work.

Also in this work we considered datasets where unlabeled data distribution doesn't significantly differ from the labeled data distribution, developing robust methods which can also take into account the distribution shift between unlabeled and labeled is an interesting direction for future work.

### A.2    Negative Societal Impact

Our work has application in fairness domain [3], where it can be used to improve performance of minority sub-groups present in data. These fairness objectives can be practically enforced on neural networks through the proposed CSST framework. However these same algorithms can be tweaked to artifically induce bias in decision making of trained neural networks, for example by ignoring performance of models on certain subgroups. Hence, we suggest deployment of these models after through testing on all sub groups of data.

## B    Connection between Minimization of Weighted Consistency Regularizer Loss (Eq. (6)) and Theoretical Weighted Consistency ($R_{\mathcal{B},w}(F)$ in Sec. 3.4)

In this section, we show that minimization of weighted consistency regularizer Eq. (6) and that of theoretical weighted consistency regularizer $R_{\mathcal{B},w}(F)$ can be related to CSL.

First, we consider $R_{\mathcal{B},w}(F)$. Using strong augmentation $\mathcal{A}$, (theoretical) weighted consistency regularizer $R_{\mathcal{B},w}(F)$ is approximated as $R_{\mathcal{B},w}(F) \approx \sum_{i,j\in[K]} w_{ij}\mathbf{E}_{x\sim P_i}\left[\mathbf{1}(F(\mathcal{A}(x)) \neq F(x))\right]$. Noting that $\mathbf{1}(F(\mathcal{A}(x)) \neq F(x)) \leq \mathbf{1}(F(\mathcal{A}(x)) \neq j) + \mathbf{1}(F(x) \neq j)$ for any $j, x$, this value is bounded as follows:

$$\begin{aligned}
R_{\mathcal{B},w}(F) &\approx \sum_{i,j\in[K]} w_{ij}\mathbf{E}_{x\sim P_i}\left[\mathbf{1}(F(\mathcal{A}(x)) \neq F(x))\right] \\
&\leq \sum_{i,j\in[K]} w_{ij}\mathbf{E}_{x\sim P_i}\left[\mathbf{1}(F(\mathcal{A}(x) \neq j))\right] + \sum_{i,j\in[K]} w_{ij}\mathbf{E}_{x\sim P_i}\left[\mathbf{1}(F(x) \neq j)\right].
\end{aligned}$$

If we focus on samples $x$ with high confidence in model predictions, then the latter term $\sum_{i,j\in[K]} w_{ij}\mathbf{E}_x\left[\mathbf{1}(F(x) \neq j)\right]$ is negligible. Therefore, minimization of (an empirical approximation of) $R_{\mathcal{B},w}(F)$ on these samples is approximately equivalent to CSL, i.e., the following problem:

$$\min_F \sum_{i,j\in[K]} w_{ij}\mathbf{E}_x\left[\mathbf{1}(F(\mathcal{A}(x)) \neq j))\right]. \tag{8}$$

The above CSL is shown to be calibrated with the loss [9] $\ell^{\mathrm{wt}}(y, p_m(x))$ (also used in Prop. 7) given below:

$$\ell^{\mathrm{wt}}(y, p_m(x)) = - \sum_{i \in [K]} G_{yi} \log \left( p_m(x)_i \right).$$

Next, we relate Eq. (6) to CSL. If we denote pseudo label $\hat{p}_m(x)$ by $y$ in Eq. (6), then we see that Eq. (6) is identical to $\ell^{\mathrm{wt}}(y, p_m(\mathcal{A}(x)))$. By (the proof of) [8, Proposition 4], by minimizing $\ell^{\mathrm{wt}}(y, p_m(\mathcal{A}(x)))$, we obtain a Bayes optimal classifier $F(\mathcal{A}(x))$, where $F$ is the classifier defined by the model $p_m$. If $w$ is the corresponding weight to the gain matrix $\mathbf{G}$, then classifier $F$ gives a solution to the CSL (8). Thus, we can relate minimization of weighted consistency regularizer Eq. (6) to that of theoretical weighted consistency regularizer $R_{\mathcal{B},w}(F)$ through the CSL (8).

# C  Additional Examples and Proof of Theorem 5

In this section, we provide some examples for assumptions introduced in Sec. 3 and a proof of Theorem 5. We provide proof of these examples in Sec. C.4.

## C.1  Examples for Theoretical Assumptions

The following example (Example 3) shows that the $c$-expansion (Definition 2) property is satisfied for mixtures of Gaussians and mixtures of manifolds.

**EXAMPLE 3.** By [14, Examples 3.4, 3.5], the $c$-expansion property is satisfied for mixtures of isotropic Gaussian distributions and mixtures of manifolds. More precisely, in the case of mixtures of isotropic Gaussian distributions, i.e., if $Q$ is given as mixtures of $\mathcal{N}(\tau_i, \frac{1}{d} I_{d \times d})$ for $i = 1, \ldots, n$ with some $n \in \mathbb{Z}_{\geq 1}$ and $\tau_i \in \mathbb{R}^d$, and $\mathcal{B}(x)$ is an $\ell_2$-ball with radius $r$ then by [1, (13)] and [14, Section B.2], $Q$ satisfies the $c$-expansion property with $c(p) = R_h(p)/p$ for $p > 0$ and $h = 2r\sqrt{d}$ (c.f., [14, section B.2]). Here $R_h(p) = \Phi(\Phi^{-1}(p) + h)$ and $\Phi$ is the cumulative distribution function of the standard normal distribution on $\mathbb{R}$.

In Sec. 3.4, we required the assumption that $\gamma > 3$ (Assumption 4) and remarked that it roughly requires $\mathrm{Err}_w(F_{\mathrm{pl}})$ is "small". The following example provides explicit conditions for $\mathrm{Err}_w(F_{\mathrm{pl}})$ that satisfy the assumption using a toy example.

**EXAMPLE 8.** Using a toy example provided in Example 3, we provide conditions that satisfy the assumption $\gamma > 3$ approximately. To explain the assumption $\gamma > 3$, we assume that $\mathcal{P}_w$ is given as a mixture of isotropic Gaussians and $\mathcal{B}(x)$ is $\ell_2$-ball with radius $r$ as in Example 3. Furthermore, we assume that $|w|_1 = 1$ and $\mathrm{Err}_w(F^*)$ is sufficiently small compared to $\mathrm{Err}_w(F_{\mathrm{pl}})$. Then, $p_w = \mathrm{Err}_w(F_{\mathrm{pl}}) + \mathrm{Err}_w(F^*) \approx \mathrm{Err}_w(F_{\mathrm{pl}})$. Using this approximation, since $\mathcal{P}_w$ satisfies the $c$-expansion property with $c(p) = R_{2r\sqrt{d}}(p)/p$, if $r = \frac{1}{2\sqrt{d}}$ then, the condition $\gamma > 3$ is satisfied when $\mathrm{Err}_w(F_{\mathrm{pl}}) < 0.17$. If $r = \frac{3}{2\sqrt{d}}$ then, the condition $\gamma > 3$ is satisfied when $\mathrm{Err}_w(F_{\mathrm{pl}}) < 0.33$.

In Assumption 1, we assumed that both of $\mathrm{Err}_w(F^*)$ and $R_{\mathcal{B},w}(F^*)$ are small. The following example suggests the validity of this assumption.

**EXAMPLE 9.** In this example, we assume $w$ is a diagonal matrix $\mathrm{diag}(w_1, \ldots, w_K)$. For simplicity, we normalize $w$ so that $\sum_{i \in [K]} w_i = 1$. As in [14, Example 3.4], we assume that $P_i$ is given as isotropic Gaussian distribution $\mathcal{N}(\tau_i, \frac{1}{d} I_{d \times d})$ with $\tau_i \in \mathbb{R}^d$ for $i = 1, \ldots, K$ and $\mathcal{B}(x)$ is an $\ell^2$-ball with radius $\frac{1}{2\sqrt{d}}$. Furthermore, we assume $\inf_{1 \leq i < j \leq K} \|\tau_i - \tau_j\|_2 \gtrsim \frac{\sqrt{\log d}}{\sqrt{d}}$ and $\sup_{i,j \in [K]} \frac{w_i}{w_j} = o(d)$, where the latter assumption is valid for high dimensional datasets (e.g., image datasets). Then it can be proved that there exists a classifier $F$ such that $R_{\mathcal{B},w}(F) = O(\frac{1}{d^c})$ and $\mathrm{Err}_w(F) = O(\frac{1}{d^c})$, where $c > 0$ is a constant (we can take $F$ as the Bayes-optimal classifier for $\mathrm{Err}_w$). Thus, this suggests that Assumption 1 is valid for datasets with high dimensional instances.

The statement of Example 8 follows from numerical computation of $R_{2r\sqrt{d}}(p)/p$. We provide proofs of Examples 3 and 9 in Sec. C.4.

 **C.2   Proof of Theorem 5 Assuming a Lemma**

104  Theorem 5 can be deduced from the following lemma (by taking $H = F^*$ and $\mathcal{L}_{Q,H}(\widehat{F}) \leq$
105  $\mathcal{L}_{Q,H}(F^*)$), which provides a similar result to [14, Lemma A.8].

106  **LEMMA 10.** *Let $H$ be a classifier and $Q$ a probability measure on $\mathcal{X}$ satisfying $c$-expansion*
107  *property. We put $\gamma_H = c(Q(\{x \in \mathcal{X} : F_{\mathrm{pl}}(x) \neq H(x)\}))$. For a classifier $F$, we define $\mathcal{S}_{\mathcal{B}}(F)$ by*
108  $\mathcal{S}_{\mathcal{B}}(F) = \{x \in \mathcal{X} : F(x) = F(x') \quad \forall x' \in \mathcal{B}(x)\}$. *For a classifier $F$, we define $\mathcal{L}_{Q,H}(F)$ by*

$$
\frac{\gamma_H + 1}{\gamma_H - 1} Q(\{x \in \mathcal{X} : F(x) \neq F_{\mathrm{pl}}(x)\})
$$
$$
+ \frac{2\gamma_H}{\gamma_H - 1} Q(\mathcal{S}_{\mathcal{B}}^c(F)) + \frac{2\gamma_H}{\gamma_H - 1} Q(\mathcal{S}_{\mathcal{B}}^c(H)) - Q(\{x \in \mathcal{X} : F_{\mathrm{pl}}(x) \neq H(x)\}),
$$

109  *where $\mathcal{S}_{\mathcal{B}}^c(F)$ denotes the complement of $\mathcal{S}_{\mathcal{B}}(F)$. Then, we have $Q(\{x \in \mathcal{X} : F(x) \neq H(x)\}) \leq$*
110  $\mathcal{L}_{Q,H}(F)$ *for any classifier $F$.*

111  In this subsection, we provide a proof of Theorem 5 assuming Lemma 10. We provide a proof of the
112  lemma in the next subsection. For a classifier $F$, we define $\mathcal{M}(F)$ as $\{x \in \mathcal{X} : F(x) \neq F^*(x)\}$ and
113  $\mathcal{M}_{\mathrm{pl}}(F)$ as $\{x \in \mathcal{X} : F(x) \neq F_{\mathrm{pl}}(x)\}$. We define $\widetilde{\mathcal{L}}_w(F)$ by

$$
\widetilde{\mathcal{L}}_w(F) = \mathcal{L}_w(F) + \frac{2\gamma}{\gamma - 1} R_{\mathcal{B},w}(F^*) - \mathcal{P}_w(\{x \in \mathcal{X} : F_{\mathrm{pl}}(x) \neq F^*(x)\}).
$$

114  We note that $\widetilde{\mathcal{L}}_w(F) - \mathcal{L}_w(F)$ does not depend on $F$.

115  *Proof of Theorem 5.* We let $Q = \mathcal{P}_w$ and $H = F^*$ in Lemma 10 and denote $\gamma_H$ in the lemma by $\gamma'$.
116  Since $w_{ij} \geq 0$ and $\mathbf{E}_{x \sim P_i}[\mathbb{1}(F_{\mathrm{pl}}(x) \neq F^*(x))] \leq \mathbf{E}_{x \sim P_i}[\mathbb{1}(F_{\mathrm{pl}}(x) \neq j)] + \mathbf{E}_{x \sim P_i}[\mathbb{1}(F^*(x) \neq j)]$
117  for any $i, j$, we have the following:

$$
|w|_1 \mathcal{P}_w(\mathcal{M}(F_{\mathrm{pl}})) = \sum_{i,j \in [K]} w_{ij} \mathbf{E}_{x \sim P_i}[\mathbb{1}(F_{\mathrm{pl}}(x) \neq F^*(x))]
$$
$$
\leq \sum_{i,j \in [K]} w_{ij} \{\mathbf{E}_{x \sim P_i}[\mathbb{1}(F_{\mathrm{pl}}(x) \neq j)] + \mathbf{E}_{x \sim P_i}[\mathbb{1}(F^*(x) \neq j)]\}
$$
$$
= \mathrm{Err}_w(F_{\mathrm{pl}}) + \mathrm{Err}_w(F^*).
$$

118  Thus, we obtain $\mathcal{P}_w(\mathcal{M}(F_{\mathrm{pl}})) \leq p_w$. Because $c$ is non-increasing, we have $\gamma \leq \gamma'$. We note that

$$
\mathrm{Err}_w(F) = \sum_{i,j \in [K]} w_{ij} \mathbf{E}_{x \sim P_i}[\mathbb{1}(F(x) \neq j)]
$$
$$
\leq \sum_{i,j \in [K]} w_{ij} \mathbf{E}_{x \sim P_i}[\mathbb{1}(F(x) \neq F^*(x))] + \sum_{i,j \in [K]} w_{ij} \mathbf{E}_{x \sim P_i}[\mathbb{1}(F^*(x) \neq j)]
$$
$$
= |w|_1 \mathcal{P}_w(\mathcal{M}(F)) + \mathrm{Err}_w(F^*).
$$

119  By this inequality and Lemma 10, the error is upper bounded as follows:

$$
\mathrm{Err}_w(F) \leq \mathrm{Err}_w(F^*) + \frac{\gamma' + 1}{\gamma' - 1} |w|_1 \mathcal{P}_w(\mathcal{M}_{\mathrm{pl}}(F))
$$
$$
+ \frac{2\gamma'}{\gamma' - 1} |w|_1 \mathcal{P}_w(\mathcal{S}_{\mathcal{B}}^c(F)) + \frac{2\gamma'}{\gamma' - 1} |w|_1 \mathcal{P}_w(\mathcal{S}_{\mathcal{B}}^c(F^*)) - |w|_1 \mathcal{P}_w(\mathcal{M}(F_{\mathrm{pl}})).
$$

120  Since $\gamma \leq \gamma'$, we obtain

$$
\mathrm{Err}_w(F) \leq \mathrm{Err}_w(F^*) + \frac{\gamma + 1}{\gamma - 1} |w|_1 \mathcal{P}_w(\mathcal{M}_{\mathrm{pl}}(F))
$$
$$
+ \frac{2\gamma}{\gamma - 1} |w|_1 \mathcal{P}_w(\mathcal{S}_{\mathcal{B}}^c(F)) + \frac{2\gamma}{\gamma - 1} |w|_1 \mathcal{P}_w(\mathcal{S}_{\mathcal{B}}^c(F^*)) - |w|_1 \mathcal{P}_w(\mathcal{M}(F_{\mathrm{pl}})). \quad (9)
$$

121  By definition of $\mathcal{L}_w$ and letting $F = \widehat{F}$, we have the following:

$$\mathrm{Err}_w(\widehat{F}) \leq \mathrm{Err}_w(F^*) + \mathcal{L}_w(\widehat{F}) + \frac{2\gamma}{\gamma - 1} R_{\mathcal{B},w}(F^*) - |w|_1 \mathcal{P}_w(\{x \in \mathcal{X} : F_{\mathrm{pl}}(x) \neq F^*(x)\})$$

$$\leq \mathrm{Err}_w(F^*) + \mathcal{L}_w(F^*) + \frac{2\gamma}{\gamma - 1} R_{\mathcal{B},w}(F^*) - |w|_1 \mathcal{P}_w(\{x \in \mathcal{X} : F_{\mathrm{pl}}(x) \neq F^*(x)\})$$

$$= \mathrm{Err}_w(F^*) + \frac{2}{\gamma - 1}|w|_1 \mathcal{P}_w(\mathcal{M}(F_{\mathrm{pl}})) + \frac{4\gamma}{\gamma - 1} R_{\mathcal{B},w}(F^*)$$

$$\leq \mathrm{Err}_w(F^*) + \frac{2}{\gamma - 1}(\mathrm{Err}_w(F_{\mathrm{pl}}) + \mathrm{Err}_w(F^*)) + \frac{4\gamma}{\gamma - 1} R_{\mathcal{B},w}(F^*)$$

$$= \frac{2}{\gamma - 1}\mathrm{Err}_w(F_{\mathrm{pl}}) + \frac{\gamma + 1}{\gamma - 1}\mathrm{Err}_w(F^*) + \frac{4\gamma}{\gamma - 1} R_{\mathcal{B},w}(F^*).$$

122  Here, the second inequality holds since $\widehat{F}$ is a minimizer of $\mathcal{L}_w$, the third inequality follows from
123  $\mathbb{1}(F^*(x) \neq F_{\mathrm{pl}}(x)) \leq \mathbb{1}(F^*(x) \neq j) + \mathbb{1}(F_{\mathrm{pl}}(x) \neq j)$ for any $j$. Thus, we have the assertion of
124  the theorem.  □

## C.3  Proof of Lemma 10

126  We decompose $\mathcal{M}(F) \cap \mathcal{S}_{\mathcal{B}}(F) \cap \mathcal{S}_{\mathcal{B}}(H)$ into the following three sets:

$$\mathcal{N}_1 = \{x \in \mathcal{S}_{\mathcal{B}}(F) \cap \mathcal{S}_{\mathcal{B}}(H) : F(x) = F_{\mathrm{pl}}(x), \text{ and } F_{\mathrm{pl}}(x) \neq H(x)\},$$
$$\mathcal{N}_2 = \{x \in \mathcal{S}_{\mathcal{B}}(F) \cap \mathcal{S}_{\mathcal{B}}(H) : F(x) \neq F_{\mathrm{pl}}(x), F_{\mathrm{pl}}(x) \neq H(x), \text{ and } F(x) \neq H(x)\},$$
$$\mathcal{N}_3 = \{x \in \mathcal{S}_{\mathcal{B}}(F) \cap \mathcal{S}_{\mathcal{B}}(H) : F(x) \neq F_{\mathrm{pl}}(x) \text{ and } F_{\mathrm{pl}}(x) = H(x)\}.$$

127  **LEMMA 11.** *Let* $S = \mathcal{S}_{\mathcal{B}}(F) \cap \mathcal{S}_{\mathcal{B}}(H)$ *and* $V = \mathcal{M}(F) \cap \mathcal{M}(F_{\mathrm{pl}}) \cap S$. *Then, we have* $\mathcal{N}(V) \cap$
128  $\mathcal{M}^c(F) \cap S = \emptyset$ *and* $\mathcal{N}(V) \cap \mathcal{M}^c(F_{\mathrm{pl}}) \cap S \subseteq \mathcal{M}_{\mathrm{pl}}(F)$. *Here* $\mathcal{M}_{\mathrm{pl}}(F)$ *is defined as* $\{x \in \mathcal{X} :$
129  $F(x) \neq F_{\mathrm{pl}}(x)\}$.

130  *Proof.* We take any element $x$ in $\mathcal{N}(V) \cap S$. Since $x \in \mathcal{N}(V)$ and definition of neighborhoods, there
131  exists $x' \in \mathcal{M}(F_{\mathrm{pl}}) \cap \mathcal{M}(F) \cap S$ such that $\mathcal{B}(x) \cap \mathcal{B}(x') \neq \emptyset$. Since $x, x' \in \mathcal{S}_{\mathcal{B}}(F)$, $F$ takes the
132  same values on $\mathcal{B}(x)$ and $\mathcal{B}(x')$. By $\mathcal{B}(x) \cap \mathcal{B}(x') \neq \emptyset$, $F$ takes the same value on $\mathcal{B}(x) \cup \mathcal{B}(x')$. It
133  follows that $F(x) = F(x')$. Since we have $x, x' \in \mathcal{S}_{\mathcal{B}}(H)$, similarly, we see that $H(x) = H(x')$. By
134  $x' \in \mathcal{M}(F)$, we have $F(x) = F(x') \neq H(x') = H(x)$. Thus, we see that $\mathcal{N}(V) \cap \mathcal{M}^c(F) \cap S = \emptyset$.
135  We assume $x \in \mathcal{N}(V) \cap \mathcal{M}^c(F_{\mathrm{pl}}) \cap S$. Then, we have $F(x) \neq H(x)$ and $F_{\mathrm{pl}}(x) = H(x)$. Therefore,
136  we obtain $F(x) \neq F_{\mathrm{pl}}(x)$. This completes the proof.  □

137  **LEMMA 12.** *Suppose that assumptions of Lemma 10 hold. We define $q$ as follows:*

$$q = \frac{Q(\mathcal{M}_{\mathrm{pl}}(F) \cup \mathcal{S}_{\mathcal{B}}^c(F) \cup \mathcal{S}_{\mathcal{B}}^c(H))}{\gamma_H - 1}. \tag{10}$$

138  *Then, we have* $Q(\mathcal{S}_{\mathcal{B}}(F) \cap \mathcal{S}_{\mathcal{B}}(H) \cap \mathcal{M}(F_{\mathrm{pl}}) \cap \mathcal{M}(F)) \leq q$. *In particular, noting that* $\mathcal{N}_1 \cup \mathcal{N}_2 \subseteq$
139  $\mathcal{S}_{\mathcal{B}}(F) \cap \mathcal{S}_{\mathcal{B}}(H) \cap \mathcal{M}(F_{\mathrm{pl}}) \cap \mathcal{M}(F)$, *we have* $Q(\mathcal{N}_1 \cup \mathcal{N}_2) \leq q$.

140  *Proof.* We let $S = \mathcal{S}_{\mathcal{B}}(F) \cap \mathcal{S}_{\mathcal{B}}(H)$ and $V = \mathcal{M}(F) \cap \mathcal{M}(F_{\mathrm{pl}}) \cap S$ as before. Then by Lemma 11,
141  we have

$$\mathcal{N}(V) \cap V^c \cap S = (\mathcal{N}(V) \cap \mathcal{M}^c(F) \cap S) \cup (\mathcal{N}(V) \cap \mathcal{M}^c(F_{\mathrm{pl}}) \cap S)$$
$$\subseteq \emptyset \cup \mathcal{M}_{\mathrm{pl}}(F) = \mathcal{M}_{\mathrm{pl}}(F).$$

142  Therefore, we have

$$\mathcal{N}(V) \cap V^c = \mathcal{N}(V) \cap V^c \cap (S \cup S^c)$$
$$= (\mathcal{N}(V) \cap V^c \cap S) \cup (\mathcal{N}(V) \cap V^c \cap S^c)$$
$$\subseteq \mathcal{M}_{\mathrm{pl}}(F) \cup S^c.$$

143  Thus, by the $c$-expansion property, we have

$$Q(\mathcal{M}_{\mathrm{pl}}(F) \cup S^c) \geq Q(\mathcal{N}(V) \cap V^c)$$
$$\geq Q(\mathcal{N}(V)) - Q(V)$$
$$\geq (c(Q(V)) - 1) Q(V).$$

144 Since $V \subseteq \mathcal{M}(F_{\mathrm{pl}})$, $c$ is non-increasing, and $\gamma_H > 1$, we have $Q(V) \le Q(\mathcal{M}_{\mathrm{pl}}(F) \cup S^c)/(\gamma_H - 1) \le q$. This completes the proof. $\qquad\square$

146 The following lemma provides an upper bound of $Q(\mathcal{N}_3)$.

147 **LEMMA 13.** *Suppose that the assumptions of Lemma 10 hold. We have*

$$Q(\mathcal{N}_3) \le q + Q(\mathcal{S}_{\mathcal{B}}^c(F) \cup \mathcal{S}_{\mathcal{B}}^c(H)) + Q(\mathcal{M}_{\mathrm{pl}}(F)) - Q(\mathcal{M}(F_{\mathrm{pl}})),$$

148 *where $q$ is defined by* (10).

149 *Proof.* We let $S = \mathcal{S}_{\mathcal{B}}(F) \cap \mathcal{S}_{\mathcal{B}}(H)$. First, we prove

$$\mathcal{N}_3 \sqcup \left(\mathcal{M}_{\mathrm{pl}}^c(F) \cap S\right) = \mathcal{N}_1 \sqcup \left(\mathcal{M}^c(F_{\mathrm{pl}}) \cap S\right). \tag{11}$$

150 Here, for sets $A, B$, we denote union $A \cup B$ by $A \sqcup B$ if the union is disjoint. By definition, we have
151 $\mathcal{N}_1 = S \cap \mathcal{M}_{\mathrm{pl}}^c(F) \cap \mathcal{M}(F_{\mathrm{pl}})$ and $\mathcal{N}_3 = S \cap \mathcal{M}_{\mathrm{pl}}(F) \cap \mathcal{M}^c(F_{\mathrm{pl}})$. Thus, we have

$$
\begin{aligned}
&\mathcal{N}_3 \cup \left(\mathcal{M}_{\mathrm{pl}}^c(F) \cap S\right) \\
&= (S \cap \mathcal{M}_{\mathrm{pl}}(F) \cap \mathcal{M}^c(F_{\mathrm{pl}})) \cup \left(\mathcal{M}_{\mathrm{pl}}^c(F) \cap S\right) \\
&= S \cap \left\{(\mathcal{M}_{\mathrm{pl}}(F) \cap \mathcal{M}^c(F_{\mathrm{pl}})) \cup \mathcal{M}_{\mathrm{pl}}^c(F)\right\} \\
&= S \cap \left(\mathcal{M}^c(F_{\mathrm{pl}}) \cup \mathcal{M}_{\mathrm{pl}}^c(F)\right).
\end{aligned}
$$

152 Similarly, we have the following:

$$
\begin{aligned}
&\mathcal{N}_1 \cup (\mathcal{M}^c(F_{\mathrm{pl}}) \cap S) \\
&= \left(S \cap \mathcal{M}_{\mathrm{pl}}^c(F) \cap \mathcal{M}(F_{\mathrm{pl}})\right) \cup (\mathcal{M}^c(F_{\mathrm{pl}}) \cap S) \\
&= S \cap \left(\mathcal{M}^c(F_{\mathrm{pl}}) \cup \mathcal{M}_{\mathrm{pl}}^c(F)\right).
\end{aligned}
$$

153 Since disjointness is obvious by definition, we obtain (11). Next, we note that the following holds:

$$
\begin{aligned}
Q(\mathcal{M}_{\mathrm{pl}}^c(F) \cap S) &= Q(\mathcal{M}_{\mathrm{pl}}^c(F)) - Q(\mathcal{M}_{\mathrm{pl}}^c(F) \cap S^c) \\
&\ge Q(\mathcal{M}_{\mathrm{pl}}^c(F)) - Q(S^c). 
\end{aligned}
\tag{12}
$$

154 By (11), we obtain the following:

$$
\begin{aligned}
Q(\mathcal{N}_3) &= Q(\mathcal{N}_1) + Q(\mathcal{M}^c(F_{\mathrm{pl}}) \cap S) - Q(\mathcal{M}_{\mathrm{pl}}^c(F) \cap S) \\
&\le Q(\mathcal{N}_1) + Q(\mathcal{M}^c(F_{\mathrm{pl}})) - Q(\mathcal{M}^c(F_{\mathrm{pl}}) \cap S) \\
&\le Q(\mathcal{N}_1) + Q(\mathcal{M}^c(F_{\mathrm{pl}})) - Q(\mathcal{M}_{\mathrm{pl}}^c(F)) + Q(S^c) \\
&\le q + Q(\mathcal{M}^c(F_{\mathrm{pl}})) - Q(\mathcal{M}_{\mathrm{pl}}^c(F)) + Q(S^c) \\
&= q - Q(\mathcal{M}(F_{\mathrm{pl}})) + Q(\mathcal{M}_{\mathrm{pl}}(F)) + Q(S^c).
\end{aligned}
$$

155 Here the second inequality follows from (12) and the third inequality follows from Lemma 12. This
156 completes the proof. $\qquad\square$

157 Now, we can prove Lemma 10 as follows.

*Proof of Lemma 10.*

$$
\begin{aligned}
Q(\mathcal{M}(F)) &= Q(\mathcal{M}(F) \cap \mathcal{S}_{\mathcal{B}}(F) \cap \mathcal{S}_{\mathcal{B}}(H)) + Q\left(\mathcal{M}(F) \cap (\mathcal{S}_{\mathcal{B}}^c(F) \cup \mathcal{S}_{\mathcal{B}}^c(H))\right) \\
&\le Q(\mathcal{N}_1 \cup \mathcal{N}_2) + Q(\mathcal{N}_3) + Q(\mathcal{S}_{\mathcal{B}}^c(F) \cup \mathcal{S}_{\mathcal{B}}^c(H)) \\
&\le 2q + 2Q(\mathcal{S}_{\mathcal{B}}^c(F) \cup \mathcal{S}_{\mathcal{B}}^c(H)) + Q(\mathcal{M}_{\mathrm{pl}}(F)) - Q(\mathcal{M}(F_{\mathrm{pl}})).
\end{aligned}
$$

158 Here, the last inequality follows from Lemmas 12 and 13. Since $q$ satisfies $q \le$
159 $\frac{Q(\mathcal{M}_{\mathrm{pl}}(F)) + Q(\mathcal{S}_{\mathcal{B}}^c(F) + Q(\mathcal{S}_{\mathcal{B}}^c(H)))}{\gamma_H - 1}$ by (10), we have our assertion. $\qquad\square$

## C.4 Miscellaneous Proofs for Examples

*Proof of Example 3.* In Example 3, we stated that $p \mapsto R_h(p)/p$ is non-increasing. This follows from the concavity of $R_h$ and $\lim_{p \to +0} R_h(p) = 0$. In fact, we can prove the concavity of $R_h$ by $\frac{d^2 R_h}{dp^2}(p) = -h \exp\left(\frac{\xi^2}{2} - h\xi - \frac{1}{2}h^2\right) \leq 0$, where $\xi = \Phi^{-1}(p)$. $\qquad\square$

*Proof of Example 9.* For each $i, j \in [K]$, $w_i P_i(x) \geq w_j P_j(x)$ is equivalent to $(x - \tau_i) \cdot v_{ji} \leq \frac{\|\tau_i - \tau_j\|}{2} + \frac{2(\log w_i - \log w_j)}{d\|\tau_i - \tau_j\|}$, where $v_{ji} = \frac{\tau_j - \tau_i}{\|\tau_i - \tau_j\|}$. Thus, for each $i \in [K]$, we have $\bigcap_{j \in [K] \setminus \{i\}} X_{ij} \subseteq S_{\mathcal{B}}(F_{\mathrm{opt}})$. Here $X_{ij}$ is defined as $\{x \in \mathcal{X} : (x - \tau_i) \cdot v_{ji} \leq \frac{\|\tau_i - \tau_j\|}{2} + \frac{2(\log w_i - \log w_j)}{d\|\tau_i - \tau_j\|} - \frac{r}{2}\}$. For any $w \in \mathbb{R}^d$ with $\|w\|_2 = \sqrt{d}$ and $a > 0$, we have $P_i(\{x \in \mathcal{X} : (x - \tau_i) \cdot w > a\}) = 1 - \Phi(a) \leq \frac{1}{2}\exp(-a^2/2)$ (c.f., [2]). Thus, $P_i(X_{ij}^c) \leq \frac{1}{2}\exp(-da_{ij}^2/2)$, where $a_{ij} = \frac{\|\tau_i - \tau_j\|}{2} + \frac{2(\log w_i - \log w_j)}{d\|\tau_i - \tau_j\|} - \frac{r}{2}$. By assumptions, we have $a_{ij}\sqrt{d} \gtrsim \sqrt{\log d}$. Therefore, $P_i(X_{ij}^c) = O(\frac{1}{poly(d)})$. It follows that $P_i(S_{\mathcal{B}}^c(F_{\mathrm{opt}})) \leq \sum_{j \in [K] \setminus \{i\}} P_i(X_{ij}^c) = O(\frac{1}{poly(d)})$. Thus, we have $R_{\mathcal{B}, w}(F_{\mathrm{opt}}) = O(\frac{1}{poly(d)})$. By the same way, we can prove that $\mathrm{Err}_w(F_{\mathrm{opt}}) = O(\frac{1}{poly(d)})$. $\qquad\square$

# D   All-Layer Margin Generalization Bounds

Following [14, 13], we introduce all layer margin of neural networks and provide generalization bounds of CSST. In this section, we assume that classifier $F(x)$ is given as $F(x) = \mathrm{argmax}_{1 \leq i \leq K} \Phi_i(x)$, where $\Phi$ is a neural network of the form

$$\Phi(x) = (f_p \circ f_{p-1} \circ \cdots \circ f_1)(x).$$

Here $f_i : \mathbb{R}^{d_i} \to \mathbb{R}^{d_{i+1}}$ with $d_1 = d$ and $d_p = K$. We assume that each $f_i$ belongs to a function class $\mathcal{F}_i \subset \mathrm{Map}(\mathbb{R}^{d_i}, \mathbb{R}^{d_{i+1}})$. We define a function class $\mathcal{F}$ to which $\Phi$ belongs by

$$\mathcal{F} = \{\Phi : \mathbb{R}^d \to \mathbb{R}^K : \Phi(x) = (f_p \circ \cdots \circ f_1)(x), \quad f_i \in \mathcal{F}_i, \forall i\}.$$

For example, for $b > 0$, $\mathcal{F}_i$ is given as $\{h \mapsto W\phi(h) : W \in \mathbb{R}^{d_i \times d_{i+1}}, \|W\|_{\mathrm{fro}} \leq b\}$ if $i > 1$ and $\{h \mapsto Wh : W \in \mathbb{R}^{d_1 \times d_2}, \|W\|_{\mathrm{fro}} \leq b\}$ if $i = 1$, where $\phi$ is a link function (applied on $\mathbb{R}^{d_i}$ entry-wise) with bounded operator norm (i.e., $\|\phi\|_{\mathrm{op}} := \sup_{x \in \mathbb{R}^{d_i} \setminus \{0\}} \|\phi(x)\|_2 / \|x\|_2 < \infty$) and $\|W\|_{\mathrm{fro}}$ denotes the Frobenius norm of the matrix. However, we do not assume the function class $\mathcal{F}_i$ does not have this specific form. We assume that each function class $\mathcal{F}_i$ is a normed vector space with norm $\|\cdot\|$. In the example above, we consider the operator norm, i.e., if $f(h) = \phi(Wh)$, $\|f\|$ is defined as $\|f\|_{\mathrm{op}}$. Let $x_1, \ldots, x_n$ be a finite i.i.d. sequence of samples drawn from $\mathcal{P}_w$. We denote the corresponding empirical distribution by $\widehat{P}_w$, i.e., for a measurable function $f$ on $\mathcal{X}$, $\mathbf{E}_{x \sim \widehat{P}_w}[f] = \sum_{i=1}^n f(x_i)$.

For $\xi = (\xi_1, \ldots, \xi_p) \in \prod_{i=1}^p \mathbb{R}^{d_i}$, we define the perturbed output $\Phi(x, \xi)$ as $\Phi(x, \xi) = h_p(x, \xi)$, where

$$h_1(x, \xi) = f_1(x) + \xi_1 \|x\|_2,$$
$$h_i(x, \xi) = f_i(h_{i-1}(x, \xi)) + \xi_i \|h_{i-1}(x, \xi)\|_2, \quad \text{for } 2 \leq i \leq p.$$

Let $x \in \mathcal{X}$ and $y \in [K]$. We define $\Xi(\Phi, x, y)$ by $\{\xi \in \prod_{i=1}^p \mathbb{R}^{d_i} : \mathrm{argmax}_i \Phi_i(x, \xi) \neq y\}$. Then, the all-layer margin $m(\Phi, x, y)$ is defined as

$$m(\Phi, x, y) = \min_{\xi \in \Xi(\Phi, x, y)} \|\xi\|_2,$$

where $\|\xi\|_2$ is given by $\sqrt{\sum_{i=1}^p \|\xi_i\|_2^2}$. Following [14], we define a variant of the all-layer margin that measures robustness of $\Phi$ with respect to input transformations defined by $\mathcal{B}(x)$ as follows:

$$m_{\mathcal{B}}(\Phi, x) := \min_{x' \in \mathcal{B}(x)} m(F, x', \mathrm{argmax}_i \Phi_i(x)).$$

**ASSUMPTION 14** (c.f. [13], Condition A.1). Let $\mathcal{G}$ be a normed space with norm $\|\cdot\|$ and $\epsilon > 0$. We say $\mathcal{G}$ satisfies the $\epsilon^{-2}$ covering condition with complexity $\mathcal{C}_{\|\cdot\|}(\mathcal{G})$ if for all $\epsilon > 0$, we have

$$\log \mathcal{N}_{\|\cdot\|}(\epsilon, \mathcal{G}) \leq \frac{\mathcal{C}_{\|\cdot\|}(\mathcal{G})}{\epsilon^2}.$$

Here $\mathcal{N}_{\|\cdot\|}(\epsilon, \mathcal{G})$ the $\epsilon$-covering number of $\mathcal{G}$. We assume function class $\mathcal{F}_i$ satisfies the $\epsilon^{-2}$ covering condition with complexity $\mathcal{C}_{\|\cdot\|}(\mathcal{F}_i)$ for each $1 \leq i \leq p$.

Throughout this section, we suppose that Assumption 14 holds. Essentially, the following two propositions follows were proved by Wei et al. [14]:

**PROPOSITION 15** (c.f., [14], Lemma D.6). *With probability at least $1 - \delta$ over the draw of the training data, for all $t \in (0, \infty)$, any $\Phi \in \mathcal{F}$ satisfies the following:*

$$R_{\mathcal{B},w}(F) = \mathbf{E}_{\widehat{P}_w}\left[\mathbf{1}(m_{\mathcal{B}}(\Phi, x) \leq t)\right] + \widetilde{O}\left(\frac{\sum_{i=1}^{p} \mathcal{C}_{\|\cdot\|_{\mathrm{op}}}(\mathcal{F}_i)}{t\sqrt{n}}\right) + \zeta,$$

*where $\zeta = O\left(\sqrt{\frac{\log(1/\delta) + \log n}{n}}\right)$ is a lower order term and $F(x) = \mathrm{argmax}_{i \in [K]} \Phi_i(x)$.*

**PROPOSITION 16** (c.f., [14], Theorem D.3). *With probability at least $1 - \delta$ over the draw of the training data, for all $t \in (0, \infty)$, any $\Phi \in \mathcal{F}$ satisfies the following:*

$$L_w(F, F_{\mathrm{pl}}) = \mathbf{E}_{\widehat{P}_w}\left[\mathbf{1}(m(\Phi, x, F_{\mathrm{pl}}(x)) \leq t)\right] + \widetilde{O}\left(\frac{\sum_{i=1}^{p} \mathcal{C}_{\|\cdot\|_{\mathrm{op}}}(\mathcal{F}_i)}{t\sqrt{n}}\right) + \zeta,$$

*where $\zeta = O\left(\sqrt{\frac{\log(1/\delta) + \log n}{n}}\right)$ is a lower order term and $F(x) = \mathrm{argmax}_{i \in [K]} \Phi_i(x)$.*

**REMARK.** Although we have proved Theorem 5 following [14], we had to provide our own proof due to some differences in theoretical assumptions (e.g., non-existence of the ground-truth classifier, a difference mentioned in the remark just after Assumption 4). On the other hand, the proofs of [14, Lemma D.6] and [14, Theorem D.3] work for any distribution $P$ on $\mathcal{X}$ and its empirical distribution $\widehat{P}$. Since $\|w\|_1 \mathcal{P}_w(\mathcal{S}_{\mathcal{B}}^c(F)) = R_{\mathcal{B},w}(F)$ and $\|w\|_1 \mathcal{P}_w(\{x : F(x) \neq F_{\mathrm{pl}}(x)\}) = L_w(F, F_{\mathrm{pl}})$, Proposition 15 and Proposition 16 follow from the corresponding results in [14].

**THEOREM 17.** *Suppose Assumption 4 and Assumption 14 hold. Then, with probability at least $1 - \delta$ over the draw of the training data, for all $t_1, t_2 \in (0, \infty)$, and any neural network $\Phi$ in $\mathcal{F}$, we have the following:*

$$\mathrm{Err}_w(F) = \frac{\gamma + 1}{\gamma - 1}\mathbf{E}_{\widehat{P}_w}\left[\mathbf{1}(m(\Phi, x, F_{\mathrm{pl}}(x)) \leq t_1)\right] + \frac{2\gamma}{\gamma - 1}\mathbf{E}_{\widehat{P}_w}\left[\mathbf{1}(m_{\mathcal{B}}(\Phi, x) \leq t_2)\right]$$

$$- \mathrm{Err}_w(F_{\mathrm{pl}}) + 2\mathrm{Err}_w(F^*) + \frac{2\gamma}{\gamma - 1}R_{\mathcal{B},w}(F^*)$$

$$+ \widetilde{O}\left(\frac{\sum_{i=1}^{p} \mathcal{C}_{\|\cdot\|_{\mathrm{op}}}(\mathcal{F}_i)}{t_1\sqrt{n}}\right) + \widetilde{O}\left(\frac{\sum_{i=1}^{p} \mathcal{C}_{\|\cdot\|_{\mathrm{op}}}(\mathcal{F}_i)}{t_2\sqrt{n}}\right) + \zeta,$$

*where $\zeta = O\left(\sqrt{\frac{\log(1/\delta) + \log n}{n}}\right)$ is a lower order term and $F(x) = \mathrm{argmax}_{i \in [K]} \Phi_i(x)$.*

*Proof.* By (9) with $H = F^*$ and $-\mathbf{1}(F_{\mathrm{pl}}(x) \neq F^*(x)) \leq -\mathbf{1}(F_{\mathrm{pl}}(x) \neq j) + \mathbf{1}(F^*(x) \neq j)$ for any $x \in \mathcal{X}$ and $j \in [K]$, we obtain the following inequality:

$$\mathrm{Err}_w(F) \leq \frac{\gamma + 1}{\gamma - 1}L_w(F, F_{\mathrm{pl}}) + \frac{2\gamma}{\gamma - 1}R_{\mathcal{B},w}(F) + \frac{2\gamma}{\gamma - 1}R_{\mathcal{B},w}(F^*) - \mathrm{Err}_w(F_{\mathrm{pl}}) + 2\mathrm{Err}_w(F^*).$$

Then, the statement of the theorem follows from Proposition 15 and Proposition 16. $\qquad\square$

## E Proof of Proposition 6

*Proof.* Let the average weighted consistency loss be $\mathcal{L}_u^{wt} = \frac{1}{|B|} \sum_{x \in B} \ell_u^{wt}(\hat{p}_m(x), p_m(\mathcal{A}(x)), \mathbf{G})$ this will be minimized if for each of $x \in B$ the $\ell_u^{wt}(\hat{p}_m(x), p_m(\mathcal{A}(x)), \mathbf{G})$ is minimized. This

expression can be expanded as:

$$\ell_u^{wt}(\hat{p}_m(x), p_m(\mathcal{A}(x)), \mathbf{G}) = -\sum_{i=1}^{K}(\mathbf{G}^T\hat{p}_m(x))_i \log(p_m(\mathcal{A}(x))_i)$$

$$= -C\sum_{i=1}^{K}\frac{(\mathbf{G}^T\hat{p}_m(x))_i}{\sum_{j=1}^{m}\mathbf{G}^T\hat{p}_m(x))_j}\log(p_m(\mathcal{A}(x))_i)$$

$$= C \times \mathrm{H}(\mathrm{norm}(\mathbf{G}^T\hat{p}_m(x)) \,||\, p_m(\mathcal{A}(x))).$$

Here we use H to denote the cross entropy between two distributions. As we don't backpropogate gradients from the $\hat{p}_m(x)$ (pseudo-label) branch of prediction network we can consider $C = \sum_{j=1}^{m}\mathbf{G}^T\hat{p}_m(x))_j$ as a constant in our analysis. Also adding a constant term of entropy $\mathrm{H}(\mathrm{norm}(\mathbf{G}^T\hat{p}_m(x)))$ to cross entropy term and dropping constant $C$ doesn't change the outcome of minimization. Hence we have the following:

$$\min_{p_m}\mathrm{H}(\mathrm{norm}(\mathbf{G}^T\hat{p}_m(x)) \,||\, p_m(\mathcal{A}(x))) = \min_{p_m}\mathrm{H}(\mathrm{norm}(\mathbf{G}^T\hat{p}_m(x)) \,||\, p_m(\mathcal{A}(x)))$$

$$+ \mathrm{H}(\mathrm{norm}(\mathbf{G}^T\hat{p}_m(x)))$$

$$= \min_{p_m}\mathcal{D}_{KL}(\mathrm{norm}(\mathbf{G}^T\hat{p}_m(x))||p_m(\mathcal{A}(x))).$$

This final term is the $\mathcal{D}_{KL}(\mathbf{G}^T\hat{p}_m(x)||p_m(\mathcal{A}(x))$ which is obtained by using the identity $\mathcal{D}_{KL}(p,q) = \mathrm{H}(p,q) + \mathrm{H}(p)$ where $p, q$ are the two distributions. $\qquad\square$

## F  Notation

We provide the list of notations commonly used in the paper in Table 1.

## G  Code, License, Assets and Computation Requirements

### G.1  Code and Licenses of Assets

In this work, we use the open source implementation of FixMatch [12] [1] in PyTorch, which is licensed under `MIT` License for educational purpose. Also for NLP experiments we make use of DistillBERT [11] pretrained model available in the HuggingFace [15] library. We promise to release the code and checkpoints at the time of acceptance of our submission.

### G.2  Computational Requirements

All experiments were done on a variety of GPUs, with primarily Nvidia A5000 (24GB) with occasional use of Nvidia A100 (80GB) and Nvidia RTX3090 (24GB). For finetuning DistilBERT and all experiments with ImageNet-100 dataset we used PyTorch data parallel over 4 A5000s. Training was done till no significant change in metrics was observed. The detailed list of computation used per experiment type and dataset have been tabulated in Table 2 and Table 3.

## H  Objective

### H.1  Logit Adjusted Weighted Consistency Regularizer

As we have introduced weighted consistency regularizer in Eq. 6 for utilizing unlabeled data, we now provide logit adjusted variant of it for training deep networks in this section. We provide logit

---

[1] https://github.com/LeeDoYup/FixMatch-pytorch

Table 1: Table of Notations used in Paper

| | | |
|---:|:---:|:---|
| $\mathcal{Y}$ | : | Label space |
| $\mathcal{X}$ | : | Instance space |
| $K$ | : | Number of classes |
| $\pi_i$ | : | prior for class i |
| $F$ | : | a classifier model |
| $s$ | : | a scoring function, $\mathcal{X} \to \mathbb{R}^K$ |
| $D$ | : | data distribution |
| $\boldsymbol{\lambda}$ | : | Lagrange multiplier |
| $\lambda_u$ | : | coefficient of unlabeled loss |
| $\mathrm{rec}_i[F]$ | : | recall of $i^{th}$ class for a classifier $F$ |
| $\mathrm{acc}[F]$ | : | accuracy for a classifier $F$ |
| $\mathrm{prec}_i[F]$ | : | precision of $i^{th}$ class for a classifier $F$ |
| $\mathrm{cov}_i[F]$ | : | coverage for $i^{th}$ class for a classifier $F$ |
| $\mathbf{G}$ | : | a $K \times K$ matrix |
| $\mathbf{D}$ | : | a $K \times K$ diagonal matrix |
| $\mathbf{M}$ | : | a $K \times K$ matrix |
| $\mu$ | : | ratio of labelled to unlabelled samples |
| $B$ | : | batch size for FixMatch |
| $\ell_u^{\mathrm{wt}}$ | : | loss for unlabelled data using pseudo label |
| $\ell_s^{\mathrm{hyb}}$ | : | loss for labelled data |
| $\mathcal{L}_u^{\mathrm{hyb}}$ | : | average loss for unlabelled data using pseudo label on a batch of samples |
| $\mathcal{L}_s^{\mathrm{hyb}}$ | : | average loss for labelled data on a batch of samples |
| $H$ | : | cross entropy function |
| $\mathcal{A}$ | : | a $\mathcal{X} \to \mathcal{X}$ function that is stochastic in nature and applies a strong augmentation to it |
| $\alpha$ | : | a $\mathcal{X} \to \mathcal{X}$ function that is stochastic in nature and applies a weak augmentation to it |
| $\rho$ | : | imbalance factor |
| $B$ | : | batch size of samples |
| $B_s$ | : | batch of labelled samples |
| $B_u$ | : | batch of unlabelled samples |
| $x$ | : | an input sample, $x \in \mathcal{X}$ |
| $\hat{p}_m$ | : | a pseudo label generating function |
| $p_m$ | : | distribution of confidence for a model's prediction on a given sample |
| $w$ | : | a $K \times K$ weight matrix that corresponds to a gain matrix $\mathbf{G}$ |
| $\mathrm{Err}_w(F)$ | : | weighted error of $F$ that corresponds to the objective of CSL |
| $\mathcal{P}_w$ | : | weighted distribution on $\mathcal{X}$ |
| $P_i$ | : | class conditional distribution of samples for class $i$ |
| $R_{\mathcal{B},w}(F)$ | : | theoretical weighted (cost sensitive) consistency regularizer |
| $F_{\mathrm{pl}}$ | : | a pseudo labeler |
| $L_w(F, F_{\mathrm{pl}})$ | : | weighted error between $F$ and $F_{\mathrm{pl}}$ |
| $\mathcal{L}_w(F)$ | : | theoretical CSST loss |
| $c$ | : | a non-increasing function used in the definition of the $c$-expansion property (Definition 2) |
| $\gamma$ | : | a value of $c$ defined in Assumption 4 |
| $\beta$ | : | an upper bound of $R_{\mathcal{B},w}(F)$ in the optimization problem (4) |
| $S^c$ | : | the complement of a set $S$ |

| Method | CIFAR-10 | CIFAR-100 | ImageNet-100 |
|---|---|---|---|
| ERM | A5000 | A5000 | RTX3090 |
| | 49m | 6h 47m | 15h 8m |
| LA | RTX3090 | A5000 | A5000 |
| | 39m | 6h 9m | 15h 7m |
| CSL | A5000 | A5000 | A5000 |
| | 47m | 6h 40m | 12h |
| CSST(FixMatch) | 4 X A5000 | 4 X A100 | 4 X A5000 |
| w/o KL-Threshold | 21h 0m | 2d 19h 16 m | 2d 13h 19m |
| CSST(FixMatch) | 4 X A5000 | 4 X A5000 | 4 X A5000 |
| | 21h 41m | 2d 11h 52m | 2d 4m |

Table 2: Computational requirements and training time (d:days, h:hours, m:minutes) for experiments relevant to vision datasets. As we can see some of the experiments on the larger datasets such as ImageNet requires long compute times.

| Method | IMDb($\rho = 10$) | IMDb($\rho = 100$) | DBpedia-14 |
|---|---|---|---|
| ERM | 4 X A5000 | 4 X A5000 | 4 X A5000 |
| | 25m | 29m | 2h 44m |
| UDA | 4 X A5000 | 4 X A5000 | 4 X A5000 |
| | 44m | 32m | 10h 18m |
| CSST(UDA) | 4 X A5000 | 4 X A5000 | 4 X A5000 |
| | 49m | 35m | 13h 12m |

Table 3: Computational requirements and training time(d:days, h:hours, m:minutes) for experiments done on NLP datasets. The DistilBERT model which we are using is pretrained on a language modeling task, hence it requires much less time for training in comparison to vision models which are trained from scratch.

adjusted term for $\ell_u^{\text{wt}}(\hat{p}_m(x), p_m(\mathcal{A}(x), \mathbf{G})$ below:

$$
\ell_u^{\text{wt}}(\hat{p}_m(x), p_m(\mathcal{A}(x), \mathbf{G}) = -\sum_{i=1}^{K} (\mathbf{G}^{\mathbf{T}}\hat{p}_m(x))_i \log(p_m(\mathcal{A}(x))_i)
$$

$$
= -\sum_{i=1}^{K} (\mathbf{G}^{\mathbf{T}}\hat{p}_m(x))_i \log \left( \frac{\exp(\mathbf{s}(\mathcal{A}(x))_i)}{\sum_{j=1}^{K} \exp(\mathbf{s}(\mathcal{A}(x))_j)} \right)
$$

$$
= -\sum_{i=1}^{K} (\mathrm{D}^{\mathbf{T}}\mathrm{M}^{\mathbf{T}}\hat{p}_m(x))_i \log \left( \frac{\exp(\mathbf{s}(\mathcal{A}(x))_i)}{\sum_{j=1}^{K} \exp(\mathbf{s}(\mathcal{A}(x))_j)} \right)
$$

The above expression comes from the decomposition $\mathbf{G} = \mathrm{MD}$. The above loss function can be converted into it's logit adjusted equivalent variant by following transformation as suggested by Narasimhan and Menon [8] which is equivalent in terms of optimisation of deep neural networks:

$$
\ell_u^{\text{wt}}(\hat{p}_m(x), p_m(\mathcal{A}(x), \mathbf{G}) \equiv -\sum_{i=1}^{K} (\mathrm{M}^{\mathbf{T}}\hat{p}_m(x))_i \log \left( \frac{\exp(\mathbf{s}(\mathcal{A}(x))_i - \log(\boldsymbol{D}_{ii}))}{\sum_{j=1}^{K} \exp(\mathbf{s}(\mathcal{A}(x))_j - \log(\boldsymbol{D}_{jj}))} \right) \quad (13)
$$

The above loss is the consistency loss $\ell_u^{\text{wt}}$ that we practically implement for `CSST`. Further in case $\hat{p}_m(x)$ is a hard pseudo label $y$ as in FixMatch, the above weighted consistency loss reduces to $\ell^{\text{hyb}}(y, \mathbf{s}(\mathcal{A}(x)))$. Further in case the gain matrix $G$ is diagonal the above loss will converge to $\ell^{\text{LA}}(y, \mathbf{s}(\mathcal{A}(x)))$. Thus the weighted consistency regularizer can be converted to logit adjusted variants $\ell^{\text{LA}}$ and $\ell^{\text{hyb}}$ based on $\mathbf{G}$ matrix.

## H.2 `CSST`(**FixMatch**)

In FixMatch, we use the prediction made by the model on a sample $x$ after applying a weak augmentation $\alpha$ and is used to get a hard pseudo label for the models prediction on a strongly

augmented sample i.e. $\mathcal{A}(x)$. The set of weak augmentations include horizontal flip, We shall refer to this pseudo label as $\hat{p}_m(x)$. The list of strong augmentations are given in Table 12 of Sohn et al. [12]. Weak augmentations include padding, random horizontal flip and cropping to the desired dimensions (32X32 for CIFAR and 224X224 for ImageNet). Given a batch of labeled and unlabeled samples $B_s$ and $B_u$, CSST modifies the supervised and un-supervised component of the loss function depending upon the non-decomposable objective and its corresponding gain matrix $\mathbf{G}$ at a given time during training. We assume that in the dataset, a sample $x$, be it labeled or unlabeled is already weakly augmented. vanilla FixMatch's supervised componenet of the loss function is a simple cross entropy loss whereas in our CSST(FixMatch) it is replaced by $\ell_s^{\text{hyb}}$ .

$$\mathcal{L}_s^{\text{hyb}} = \frac{1}{|B_s|} \sum_{x,y \in B_s} \ell^{\text{hyb}}(y, s(x)). \tag{14}$$

$$\mathcal{L}_u^{\text{wt}} = \frac{1}{|B_u|} \sum_{x \in B_u} \mathbb{1}_{(\mathcal{D}_{KL}(\text{norm}(\mathbf{G}^T \hat{p}_m(x)) \, || \, p_m(x)) \leq \tau)} \ell_u^{\text{wt}}(\hat{p}_m(x), p_m(\mathcal{A}(x)), \mathbf{G})). \tag{15}$$

In the above expression $p_m(x) = \text{softmax } \mathbf{s}(x)$. The component of the loss function for unlabeled data (i.e. consistency regularization) is where one of our contributions w.r.t the novel thresholding mechanism comes into light. vanilla FixMatch selects unlabeled samples for which consistency loss is non-zero, such that the model's confidence on the most likely predicted class is above a certain threshold. We rather go for a threshold mechanism that select based on the basis of degree of distribution match to a target distribution based on $\mathbf{G}$. The final loss function $\mathcal{L} = \mathcal{L}_s^{hyb} + \lambda_u \mathcal{L}_u^{\text{wt}}$, i.e. a linear combination of $\mathcal{L}_s^{\text{hyb}}$ and $\mathcal{L}_u^{wt}$. Since for FixMatch we are dealing with Wide-ResNets and ResNets which are deep networks, as mentioned in Section H.1, we shall use the alternate logit adjusted formulation as mentioned in Eq. 13 as substitute for $\ell_u^{\text{wt}}$ in Eq. 15.

**H.3  CSST(UDA)**

The loss function of UDA is a linear combination of supervised loss and consistency loss on unlabeled samples. The former is the cross entropy (CE) loss, while the latter for the unlabeled samples minimizes the KL-divergence between the model's predicted label distribution on an input sample and its augmented sample. Often the predicted label distribution on the unaugmented sample is sharpened. The augmentation we used was a English-French-English backtranslation based on the MarianMT [4] fast neural machine translation model. In UDA supervised component of the loss is annealed using a method described as Training Signal Annealing (TSA), where the CE loss is considered only for those labeled samples whose $\max_i p_m(x)_i < \tau_t$, where $t$ is a training time step. We observed that using TSA in a long tailed setting leads to overfitting on the head classes and hence chose to not include the same in our final implementation.

CSST modifies the supervised and unsupervised component of the loss function in UDA depending upon a given objective and its corresponding gain matrix $\mathbf{G}$ at a given time during training. The supervised component of the loss function for a given constrained optimisation problem and a gain matrix $\mathbf{G}$, is the hybrid loss $\ell_s^{\text{hyb}}$. For the consistency regularizer part of the loss function, we minimize the KL-divergence between a target distribution and the model's prediction label distribution on its augmented version. The target distribution is $\text{norm}(\mathbf{G}^T \hat{p}_m(x))$, where $\hat{p}_m(x)$ is the sharpened prediction of the label distribution by the model. Given a batch of labeled and unlabeled samples $B_s$ and $B_u$, the final loss function in CSST(UDA) is a linear combination of $\mathcal{L}_s^{hyb}$ and $\mathcal{L}_u^{wt}$, i.e $\mathcal{L} = \mathcal{L}_s^{hyb} + \lambda_u \mathcal{L}_u^{wt}$.

$$\mathcal{L}_s^{\text{hyb}} = \frac{1}{|B_s|} \sum_{x,y \in B_s} \ell^{\text{hyb}}(p_m(x), y). \tag{16}$$

$$\mathcal{L}_u^{wt} = \frac{1}{|B_u|} \sum_{x \in B_u} \mathbb{1}_{(\mathcal{D}_{KL}(\text{norm}(\mathbf{G}^T \hat{p}_m(x)) \, || \, p_m(x)) \leq \tau)} \ell_u^{\text{wt}}(\hat{p}_m(x), p_m(\mathcal{A}(x), \mathbf{G})). \tag{17}$$

Since for UDA, we are dealing with DistilBERT, as mentioned in Section H.1, we shall use the alternate formulation as mentioned in Eq. 13 as substitute for $\ell_u^{\text{wt}}$ in Eq. above.

# I Threshold mechanism for diagonal Gain Matrix

Consider the case when the gain matrix is a diagonal matrix. The loss function $\mathcal{L}_u^{wt}(B_u)$ as defined in (7) makes uses of a threshold function that selects samples based on the KL divergence based threshold between the target distribution as defined by the gain matrix $\mathbf{G}$ and the models predicted distribution of confidence over the classes.

$$\text{Threshold function} := \mathbb{1}_{\left(\mathcal{D}_{KL}\left(\text{norm}(\mathbf{G}^T \hat{p}_m(x)) \,\|\, p_m(x)\right) \leq \tau\right)} \tag{18}$$

Since $\mathbf{G}$ is a diagonal matrix and the pseudo-label $\hat{p}_m(x)$ is one hot, the $\text{norm}(\mathbf{G}^T \hat{p}_m(x))$ is a one-hot vector. The threshold function's KL divergence based criterion can be expanded as follows where $\hat{y}$ is the pseudo-label's maximum class's index:

$$\mathcal{D}_{KL}(\text{norm}(\mathbf{G}^T \hat{p}_m(x)) \| p_m(x)) = -\log p_m(x)_{\hat{y}} < \tau \tag{19}$$

The above equations represents a threshold on the negative log-confidence of the model's prediction for a given unlabeled sample, for the pseudo-label class ($\hat{y}$). This can be further simplified to $p_m(x)_{\hat{y}} \geq \exp(-\tau)$ which is simply a threshold based on the model's confidence. Since pseudo-label is generated from the model's prediction, this threshold is nothing but a selection criterion to select only those samples whose maximum confidence for a predicted hard pseudo-label is above a fixed threshold. This is identical to the threshold function which is used in Fixmatch [12] i.e. $\max(p_m(x)) \geq \exp(-\tau)$. In FixMatch this $\exp(-\tau)$ is set to 0.95.

# J Dataset

**CIFAR-10 and CIFAR-100 [5].** are image classification datasets of images of size 32 X 32. Both the datasets have a size of 50k samples and by default, they have a uniform sample distribution among its classes. CIFAR-10 has 10 classes while CIFAR-100 has 100 classes. The test set is a balanced set of 10k images.

**ImageNet-100 [10].** is an image classification dataset carved out of ImageNet-1k by selecting the first 100 classes. The distribution of samples is uniform with 1.3k samples per class. The test set contains 50 images per class. All have a resolution of 224X224, the same as the original ImageNet-1k dataset.

**IMDb[7].** dataset is a binary text sentiment classification dataset. The data distribution is uniform by default and has a total 25k samples in both trainset and testset. In this work, we converted the dataset into a longtailed version of $\rho = 10, 100$ and selected 1k labeled samples while truncating the labels of the rest and using them as unlabeled samples.

**DBpedia-14[6].** is a topic classification dataset with a uniform distribution of labeled samples. The dataset has 14 classes and has a total of 560k samples in the trainset and 70k samples in the test set. Each sample, apart from the content, also has title of the article that could be used for the task of topic classification. In our experiments, we only make use of the content.

# K Algorithms

We provide a detailed description of algorithms used for optimizing non decomposable objectives through CSST(FixMatch) ans CSST(UDA). Algorithm 1 is used for experiments in Section 5 for maximizing worst-case recall (i.e. min recall using CSST(FixMatch) and CSST(UDA)). Algorithm 2 is used for experiments in Section 5 for maximizing recall under coverage constraints (i.e. min coverage experiments on CIFAR10-LT, CIFAR100-LT and ImageNet100-LT).

---

**Algorithm 1** CSST-based Algorithm for Maximizing Worst-case Recall

---

Inputs: Training set $S_s$(labeled) and $S_u$(unlabeled) , Validation set $S^{\text{val}}$, Step-size $\omega \in \mathbb{R}_+$, Class priors $\pi$
Initialize: Classifier $h^0$, Multipliers $\boldsymbol{\lambda}^0 \in \Delta_{K-1}$
**for** $t = 0$ to $T - 1$ **do**
    **Update $\boldsymbol{\lambda}$:**
      $\lambda_i^{t+1} = \lambda_i^t \exp\left(-\omega \cdot \text{recall}_i[F^t]\right), \forall i,$
      $\boldsymbol{\lambda} = \text{norm}(\boldsymbol{\lambda})$
      $\mathbf{G} = \text{diag}(\lambda_1^{t+1}/\pi_1, \ldots, \lambda_K^{t+1}/\pi_K)$
    Compute $\ell_u^{\text{wt}}, \ell_s^{\text{hyb}}$ using $\mathbf{G}$
    **Cost-sensitive Learning (CSL) for FixMatch:**
      $B_u \sim S_u, B_s \sim S_s$ // Sample batches of data
      $F^{t+1} \in \arg\min_F \sum_{B_u, B_s} \lambda_u \mathcal{L}_u^{\text{wt}} + \mathcal{L}_s^{\text{hyb}}$ // Replaced by few steps of SGD
**end for**
**return** $F^T$

---

---

**Algorithm 2** CSST-based Algorithm for Maximizing Mean Recall s.t. per class coverage > 0.95/K

---

Inputs: Training set $S_s$(labeled) and $S_u$(unlabeled) , Validation set $S^{\text{val}}$, Step-size $\omega \in \mathbb{R}_+$, Class priors $\pi$
Initialize: Classifier $h^0$, Multipliers $\boldsymbol{\lambda}^0 \in \mathbb{R}_+^K$
**for** $t = 0$ to $T - 1$ **do**
    **Update $\boldsymbol{\lambda}$:**
      $\lambda_i^{t+1} = \lambda_i^t - \omega\left(\text{cov}_i[F^t] - \frac{0.95}{K}\right), \forall i$
      $\lambda_i^{t+1} = \max\{0, \lambda_i^{t+1}\}, \forall i \in [K]$    // Projection to $\mathbb{R}_+$
      $\mathbf{G} = \text{diag}(\lambda_1^{t+1}/\pi_1, \ldots, \lambda_K^{t+1}/\pi_K) + \mathbf{1_K}\boldsymbol{\lambda}^\top$
    Compute $\ell_u^{\text{wt}}, \ell_s^{\text{hyb}}$ using $\mathbf{G}$
    **Cost-sensitive Learning (CSL) for FixMatch:**
      $B_u \sim S_u, B_s \sim S_s$ // Sample batches of data
      $F^{t+1} \in \arg\min_F \sum_{B_u, B_s} \lambda_u \mathcal{L}_u^{\text{wt}} + \mathcal{L}_s^{\text{hyb}}$ // Replaced by few steps of SGD
**end for**
**return** $F^T$

---

## L   Details of Experiments and Hyper-parameters

The experiment of $\max_F \min_i \text{recall}_i[f]$ and $\max_F \text{recall}[F]$ s.t. $\text{cov}_i[F] > \frac{0.95}{K}, \forall i \in [K]$ was performed on the long tailed version of CIFAR-10, IMDb($\rho = 10, 100$) and DBpedia-14 datasets. This was because the optimisation of the aforementioned 2 objectives is stable for cases with low number of classes. Hence the objective of $\max_F \min(\text{recall}_{\mathcal{H}}[F], \text{recall}_{\mathcal{T}}[F])$ and $\max_F \text{recall}[F]$ s.t. $\min_{\mathcal{H},\mathcal{T}} \text{cov}_{\mathcal{H},\mathcal{T}}[F] > \frac{0.95}{K}$ is a relatively easier objective for datasets with large number of classes, hence were the optimisation objectives for CIFAR-100 and ImageNet-100 long tailed datasets. For all experiments for a given dataset, we used the same values for a given common hyperparameter. We ablated the threshold for our novel unlabeled sample selection criterion($\tau$) and the ratio of labeled and unlabeled samples, given fixed number of unlabeled samples($\mu$) and are available in Fig. 4b.

## M   Statistical Analysis

We establish the statistical soundness and validity of our results we ran our experiments on 3 different seeds. Due to the computational requirements for some of the experiments ($\approx$2days) we chose to run the experiments on multiple seeds for a subset of tasks i.e. for maximising the minimum recall among all classes for CIFAR-10 LT. We observe that the std. deviation is significantly smaller than the average values for mean recall and min. recall and our performance metrics fall within our std. deviation hence validating the stability and soundness of training.

| Parameter | CIFAR-10 | CIFAR-100 | ImageNet-100 | IMDb ($\rho = 10$) | IMDb ($\rho = 100$) | DBpedia-14 |
|---|---|---|---|---|---|---|
| $\tau$ | 0.05 | 0.05 | 0.05 | 0.1 | 0.1 | 0.1 |
| $\lambda_u$ | 1.0 | 1.0 | 1.0 | 0.1 | 0.1 | 0.1 |
| $\mu$ | 4.0 | 4.0 | 4.0 | 13.8 | 12.6 | 133 |
| $|B_s|$ | 64 | 64 | 64 | 32 | 32 | 32 |
| $|B_u|$ | 256 | 256 | 256 | 128 | 128 | 128 |
| lr | 3e-3 | 3e-3 | 0.1 | 1e-5 | 1e-5 | 1e-5 |
| $\omega$ | 0.25 | 0.25 | 0.1 | 0.5 | 0.5 | 0.5 |
| SGD steps before eval | 32 | 100 | 500 | 50 | 50 | 100 |
| optimizer | SGD | SGD | SGD | AdamW | AdamW | AdamW |
| KL-Thresh | 0.95 | 0.95 | 0.95 | 0.9 | 0.9 | 0.9 |
| Weight Decay | 1e-4 | 1e-3 | 1e-4 | 1e-2 | 1e-2 | 1e-2 |
| $\rho$ | 100 | 10 | 10 | 10 | 100 | 100 |
| $\lambda_u$ | 1.0 | 1.0 | 1.0 | 0.1 | 0.1 | 0.1 |
| Arch. | WRN-28-2 | WRN-28-8 | ResNet50 | DistilBERT | DistilBERT | DistilBERT |

Table 4: This table shows us the detailed hyper parameters used for CSST(FixMatch) for the long tailed datasets CIFAR-10, CIFAR-100, ImageNet-100 and CSST(UDA) on IMDb, DBpedia-14. All the datasets were converted to their respective long tailed versions based on the imbalance factor $\rho$, and a fraction of the samples were used along with their labels for supervision.

Table 5: Avg. and std. deviation of Mean Recall and Min. Recall for CIFAR-10 LT

| Method | Mean Recall | Min Recall |
|---|---|---|
| ERM | $0.52 \pm 0.01$ | $0.27 \pm 0.02$ |
| LA | $0.54 \pm 0.02$ | $0.37 \pm 0.01$ |
| CSL | $0.63 \pm 0.01$ | $0.43 \pm 0.04$ |
| Vanilla (FixMatch) | $0.78 \pm 0.01$ | $0.47 \pm 0.02$ |
| CSST(FixMatch) | $0.75 \pm 0.01$ | $0.72 \pm 0.01$ |

## N    Additional Details

### N.1    Formal Statement Omitted in Sec. 2.2

In Sec. 2.2, we stated that learning with the hybrid loss $\ell^{\text{hyb}}$ gives the Bayes-optimal classifier for the CSL (2). However, due to space constraint, we did not provide a formal statement. In this section, we provide a formal statement of it for clarity.

**PROPOSITION** ([8] Proposition 4). For any diagonal matrix $\mathbf{D} \in \mathbb{R}^{K \times K}$ with $D_{ii} > 0, \forall i$, $\mathbf{M} \in \mathbb{R}^{K \times K}$, and $\mathbf{G} = \mathbf{M}\mathbf{D}$, the hybrid loss $\ell^{\text{hyb}}$ is calibrated for $\mathbf{G}$. That is, for any score function $\widehat{\mathbf{s}} : \mathcal{X} \to \mathbb{R}^K$ that minimizes $\mathbf{E}_{(x,y) \sim D}\left[\ell^{\text{hyb}}(y, \mathbf{s}(x))\right]$, the associated classifier $F(x) = \operatorname{argmax}_{y \in [K]} \widehat{s}_i(x)$ is the Bayes-optimal classifier for CSL (2).

### N.2    Comparison with the $(a, \widetilde{c})$-expansion Property in [14]

We compare the $c$-expansion property with $(a, \widetilde{c})$-expansion property proposed by [14], where $a \in (0, 1)$ and $\widetilde{c} > 1$. Here we say a distribution $Q$ on $\mathcal{X}$ satisfies the $(a, \widetilde{c})$-expansion property if $Q(\mathcal{N}(S)) \geq \widetilde{c}$ for any $S \subset \mathcal{X}$ with $Q(S) \leq a$. If $Q$ satisfies $(a, \widetilde{c})$-expansion property [14] with $\widetilde{c} > 1$, then $Q$ satisfies the $c$-expansion property, where the function $c$ is defined as follows. $c(p) = \widetilde{c}$ if $p \leq a$ and $c(p) = 1$ otherwise. On the other hand, if $Q$-satisfies $c$-expansion property, then for any $a \in (0, 1)$ and $S \subseteq \mathcal{X}$ with $Q(S) \leq a$, we have $Q(\mathcal{N}(S)) \geq c(Q(S))Q(S) \geq c(a)Q(S)$ since $c$ is non-increasing. Therefore, $Q$ satisfies the $(a, c(a))$-expansion property. Thus, we could say these two conditions are equivalent. To simplify our analysis, we use our definition of the expansion property.

In addition, Wei et al. [14] showed that the $(a, \widetilde{c})$-expansion property is realistic for vision. Although they assumed the $(a, c)$-expansion property for each $P_i$ ($1 \leq i \leq K$) and we assume the $c$-expansion property for $\mathcal{P}_w$, it follows that the $c$-expansion property for $\mathcal{P}_w$ is also realistic for vision, since $\mathcal{P}_w$ is a linear combination of $P_i$.

## N.3 Comparison of the Theoretical Assumptions with that of [14]

In the last paragraph of Sec. 3.1, we explain the difference in the theoretical assumptions between ours and [14]. Wei et al. [14] assumed the existence of the ground-truth classifier and the supports of $P_i$ are disjoint, but we cannot assume these conditions due to our problem setting (i.e., optimizing the cost sensitive objectives). In this section, we provide more intuitive explanation using a toy example. For simplicity, we assume $K = 2$, $\mathcal{X} \subset \mathbb{R}$ and $w = \mathrm{diag}(w_1, w_2)$ with $w_1, w_2 \geq 0$.

In Fig. 1, we consider the cost sensitive (weighted) objective in the case where supports of $P_1$ and $P_2$ are disjoint. As the figure indicates the Bayes optimal classifier $x \mapsto \mathrm{argmax}_{i \in [K]} w_i P_i(x)$ for the cost sensitive objective does not depend on $w$. The ground truth classifier (i.e., $x \mapsto \mathrm{argmax}_{i \in [K]} P_i(x)$) is the best classifier for any $w$.

On the other hand, in Fig. 2, we consider a more generalized setting where the supports are not necessarily disjoint. In this case, the optimal classifier for the cost sensitive objective depends on $w$. This simple example suggests that we have to generalize [14] by removing the restrictive assumptions on the supports and the ground truth classifier.

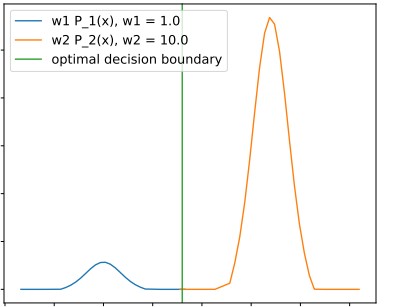
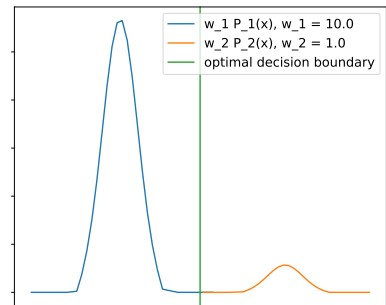

Figure 1: In the perfect setting where two distributions have disjoint supports, the Bayes optimal classifier for the CSL is identical to the ground truth classifier ($x \mapsto \mathrm{argmax}_i P_i(x)$) for any choices of weights $(w_1, w_2)$.

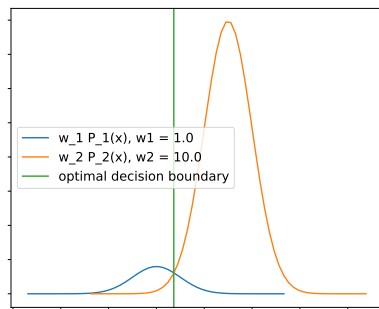
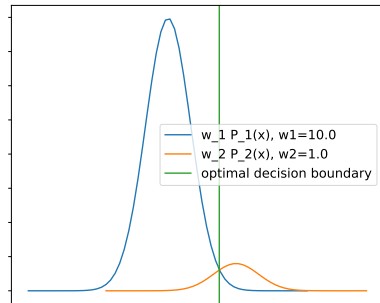

Figure 2: In more generalized settings, the Bayes optimal classifier for the CSL depends on the choice of weights (i.e., gain matrix). In the left figure, we put more weight on the second class than the first class. In the right figure, we put less weight on the second class than the first class. By decreasing the weight $w_2$, the optimal decision boundary for the CSL moves to the right.