# OpenReview forum: "Cost-Sensitive Self-Training for Optimizing Non-Decomposable Metrics"
_NeurIPS.cc/2022/Conference — NeurIPS 2022 Accept_

### Official Review · Reviewer_GPmX · 2022-07-08

**Rating:** 8
**Confidence:** 5
**Soundness:** 4 excellent
**Presentation:** 3 good
**Contribution:** 4 excellent

**Summary:**

Non-decomposable metrics take an important role in machine learning. This paper proposes to improve self-training by optimizing non-decomposable objectives for semi-supervised learning. This strategy can bring a significant average improvement in desired metric of minimizing worst-case recall while maintaining similar accuracy compared with SOTA methods. Empirical results show that proposed Cost-Sensitive Self-Training (CSST) framework help improve the performace of baselines, e.g., FixMatch, UDA.

**Questions:**

See above.

**Ethics Review Area:**

["I don’t know"]

**Limitations:**

Yes.

**Strengths And Weaknesses:**

Pros
- Propose to improve self-training method by optimizing non-decomposable metrics that utilizes unlabeled data in addition to labeled data.
- Propose a weighted consistency regularizer for Cost-Sensitive Self-Training.
- The superiority of desired non-decomposable metric is theoretically justified.
- Empirical results have validated the effectness of the proposed method, and bring improvement over baselines.


I am overall satisfied with this paper and would like to give some comments that may make this paper better.
- It is expected to release the source code of all experiments to benifit the community.
- It is promising to compare with methods that improves the recall of Fixmatch, e.g., Flexmatch.

---

> ### Author Response · Authors · 2022-08-02
> **Response to reviewer GPmX**
>
> Thank you for your suggestions and encouraging comments.
>
> #### **Response to Suggestion 1**
> We shall make the codebase available immediately after the acceptance of our paper for the benefit of the community.
>
> #### **Response to Suggestion 2**
> We looked into previous works for Long-Tail learning in a Semi-Supervised Learning setting and compared our results for the target objectives of maximising the worst-case recall and also maximising the mean recall subject to a target coverage constraint. We compared against CReST[1], and DARP[2]. As suggested we also compared the results against FlexMatch[3].  We observed that CSST(FixMatch) (Ours) gave the best results for the given target objective in all the cases.
>
> **Table**: Maximising minimum recall for CIFAR10 and minimum of Head and Tail recall for CIFAR-100
>
> | Method         |       CIFAR-10  (Imbalance=100) |       |      CIFAR-100  (Imbalance=10)     |      |
> |:----------------:|:-------------:|:--------------:|:-------------:|:------------------:|
> |                | Mean Recall | Min Recall | Mean Recall | Min H-T Recall |
> | CReST[1]          | 0.72        | 0.47       | 0.52          | 0.46           |
> | DARP[2]           | 0.81        | 0.64       | 0.55          | 0.54           |
> | FlexMatch[3]           | 0.80        | 0.48       | 0.61          | 0.39           |
> | CSST(FixMatch)[4] | 0.76        | 0.72       | 0.63          | 0.61           |
>
>
>
> **Table**: Maximising Mean Recall with a target (tgt.) coverage constraint
>
>
> | Method         |       CIFAR-10  (Imbalance=100) |       |      CIFAR-100  (Imbalance=10)     |      |
> |:----------------:|:-------------:|:--------------:|:-------------:|:------------------:|
> |                | Mean Recall | Min Coverage | Mean Recall | Min H-T Coverage|
> |                |  | (tgt.  0.095) | |  (tgt. 0.01)|
> | CReST [1]         | 0.72        | 0.052        | 0.52          | 0.009            |
> | DARP   [2]        | 0.81        | 0.063        | 0.55          | 0.006            |
> | FlexMatch[3]           | 0.80        | 0.046       | 0.61          | 0.006           |
> | CSST(FixMatch) [4]| 0.80        | 0.092        | 0.63          | 0.010            |
>
>
> [1]: Chen Wei, Kihyuk Sohn, Clayton Mellina, Alan Yuille, Fan Yang, CReST: A Class-Rebalancing Self-Training Framework for Imbalanced Semi-Supervised Learning, CVPR '21 \
> [2]: Jaehyung Kim, Youngbum Hur, Sejun Park, Eunho Yang, Sung Ju Hwang, Jinwoo Shin, Distribution Aligning Refinery of Pseudo-label for Imbalanced Semi-supervised Learning NeurIPS '20 \
> [3]: Bowen Zhang, Yidong Wang, Wenxin Hou, Hao Wu, Jindong Wang, Manabu Okumura, Takahiro Shinozaki
> ,FlexMatch: Boosting Semi-Supervised Learning with Curriculum Pseudo Labeling NeurIPS '21\
> [4]: Cost-Sensitive Self-Training for Optimizing Non-Decomposable Metrics (Ours)

---

### Official Review · Reviewer_FQ9c · 2022-07-11

**Rating:** 3
**Confidence:** 3
**Soundness:** 2 fair
**Presentation:** 1 poor
**Contribution:** 2 fair

**Summary:**

The paper propose a new loss function for self learning that can take into account also non-decomposable cost functions. The work both analyzes the performance when using this new cost function and show empirically its advantage over previous solutions for self-training. In the theoretical part, the author claim that their loss function leads to a lower error compared to standard optimization over the loss. In the practical part, the authors show that their approach, which combines also a smart thresholding for keeping only part of the examples for the self training, leads to improved results over previous solutions. Both vision and NLP benchmarks are being considered.

**Questions:**

see above

**Limitations:**

Didn't see limitations being mentioned

**Strengths And Weaknesses:**

The strength of the paper is that it combines a theoretical analysis with some strong empirical results compared to existing baselines.
A new loss function is being proposed and a thresholding approach. Both seems to improve the performance for the task at hand

Weaknesses:

1. The paper is very (very very) hard to read. The notation are not well presented. Some symbols are defined only after they are being first used. The writing is quite bad and it is very hard to follow. Few examples:
a.  the recall and precision on line 65-66 are defined the same.
b. F_{pl =} is used in Assumption 1 but defined only in Section 3.4
c. The notation is also not very consistent. Sometimes the authors use P_w where w is the weight and in the same equation use P_i where i is an index. It is very unclear and confusing.
d. Less critical compared to the above but part of the things that are defined should be better defined. For example c that is used in assumption 4 is mentioned without saying what it is. Indeed, it is mentioned in the c-expansion but the statement should start with mentioning that there is a function c and then say what is done with it. The comment here, is indeed a matter of style (unlike point a-c, which are not a matter of style but just make the paper very hard to follow)

2. The assumption made in theoretical part are not justified. Specifically, in assumption 1 if we assume beta is negligible then it means that R_B,w is basically equal to zero. Is it really the constraint we want to use?
Also, if we assume that the error of the found function F^* is smaller than the one of F_{pl}, why Theorem 5 is surprising? Is not it just saying that if the error is smaller than the error is smaller? The authors assume something and then claim something more complicated that more or less states the same. At the end of the day, they want to show that F^* is better. So they assume that and then claim they proved that. So what is the point?

3. The experiments are interesting but I wonder whether the main benefit is just from the new thresholding being used. There is an ablation that show the impact of this new threshoding approach on the proposed CSST approach. Yet, the real missing experiment is checking what is the impact of this approach on existing baselines (e.g., FixMatch). Clearly, the problem with the threshold has already been mentioned in previous works as the authors admit. So proposing a solution for it should be checked also with previous works.

4. Finally, why only one baseline is being compared to in each task?  Is FixMatch/UDA the only work that was proposed sine 2020 for this problem (Clearly there are and the authors should compare to them). Note that this is the least important problem in the paper that I find compared to the other points raised above.

---

> ### Author Response · Authors · 2022-08-03
> **Response to Reviewer FQ9c (1/3)**
>
> We sincerely thank the reviewer for their insightful and critical comments. We believe that the major misunderstanding has been considering $F^\star$ to be the found classifier, whereas we use $F^\star$ to denote the optimal classifier and $\hat{F}$ to be found classifier in Theorem 5. We have put *significant effort into improving the clarity of the draft and would be grateful if you can please have a look at the revised version*.
>
> ### Response to 1
>
> > Some symbols are defined only after they are being first used.
>
> We apologize for the inconvenience. We have now fixed the notation of precision and recall, defined $F_{pl}$ before Assumption 1. We also would like to mention that we have provided a table of notations in the Appendix for reference.
>
> >Sometimes the authors use P_w where w is the weight and in the same equation use P_i where i is an index.
>
> The ${P_w}$ is a drop in replacement for $P_i$ in our framework, as it's the weighted average of $P_i$ defined as: $P_w = \sum_{i,j} w_{i,j}P_i$. Hence, $P_{w}$ doesn't have any index $i$. Furthermore, now we use $\mathcal{P}_w$ in place of ${P_w}$  to denote the weighted probability to avoid confusion.
>
>
> > Effort on improving Clarity
>
> We have now improved the clarity by introducing a summary Table for metrics, improving notational consistency, changing definition style, and providing a formal statement of each theorem used from prior literature in the appendix. A detailed list of changes has been specified in the summary of revision that we hope would significantly aid the readability of the paper and improves its clarity. We sincerely thank you for your suggestions.
>
> ### Response to 2.
>
> > Weakness 2: Specifically, in assumption 1 if we assume $\beta$ is negligible then it means that R_B,w is basically equal to zero. Is it really the constraint we want to use?
>
>
> - The condition $R_{B, w}(F)$ being small implies that $F$ is robust to data augmentation. Therefore, it is expected that a good classifier $F$ has a low value of weighted consistency $R_{B, w}(F)$. Since $\beta$ is an upper bound of $R_{B, w}(F^\star)$  and $F^\*$ is an optimal classifier, we think even though $\beta$ is small, it's natural to assume such an $F^\*$ exists as we use overparameterized neural network based classifiers and high-dimensional (d) data (e.g., image data, etc.).
> - In addition, as we remarked just after Assumption 1, we also provide an example that justifies the existence of optimal $F^\star$ under this assumption using a simple and common data generation model too, in Appendix C.1, Example 9. We show that for optimal classifier $F^\star$ the optimal weighted consistency $R_{B,w} (F^\star)$ is $O(\frac{1}{poly(d)})$, which is negligible for large $d$ (here $d$ is the dimension of input i.e. large for image data).
> - Our assumption on $\beta$ to be negligible is very similar to the assumption (3.3) made in Wei et al. [38] for theoretical analysis of self-training algorithms like FixMatch, which also justifies its applicability here. Hence, we also use the constraint of $\beta$ to be small in our work.
>
> > Weakness 2: Also, if we assume that the error of the found function F^* is smaller than the one of F_{pl}, why Theorem 5 is surprising?
>
> - We would like to clarify that it's classifier **$\hat{F}$ (not $F^\*$)** that is found by minimizing the loss in Eq. 5. $F^\*$ instead is the **optimal classifier** that minimizes the $Err_{w}(F)$ subject to the weighted consistency constraint $R_{B,w}(F) \leq \beta$ (i.e. robust to data augmentation) defined in Eq. 4, which is **unknown** for us.
> - The statement of Theorem 5 is non-trivial (and surprising) since the learned classifier $\hat{F}$ using the loss function in Eq. 5 has superior performance than the pseudo labeler $F_{pl}$ in terms of the cost-sensitive metric, and the loss function for $\hat{F}$  is defined using only pseudo labeler $F_{pl}$ and the weighted consistency regularizer $R_{B, w}$ which does not require labels.
>
> > At the end of the day, they want to show that F^* is better. So they assume that and then claim they proved that. So what is the point?
> - We want to convey that the learnt classifier **$\hat{F}$ (not $F^\*$)** is better than $F_{pl}$. *Obtaining a better classifier from lower performing pseudo labeler and regularization, justified theoretically through Theorem 5, explains the success of our CSST method.*

---

> > ### Author Response · Authors · 2022-08-03
> > **Response to Reviewer FQ9c (2/3)**
> >
> > ### Response to 3.
> >
> > > Yet, the real missing experiment is checking what is the impact of this approach on existing baselines.
> >
> > For investigating this we run the FixMatch algorithm with the proposed KL-Thresholding method in CSST for the objective of maximizing mean recall under coverage constraints (Section 5). We tabulate the results below.
> >
> >
> > | Method         |       CIFAR-10  (Imbalance=100) |       |      CIFAR-100  (Imbalance=10)     |      |
> > |:----------------:|:-------------:|:--------------:|:-------------:|:------------------:|
> > |                | Mean Recall | Min Coverage | Mean Recall | Min H-T Coverage|
> > |                |  | (tgt.  0.095) | |  (tgt. 0.01)|
> > | CSST(FixMatch) w/o  weighted consistency regularizer           | 0.55        | 0.017       | 0.44          | 0.004           |
> > | CSST(FixMatch) [4]| 0.80        | 0.092        | 0.63          | 0.010            |
> >
> > We find that this leads to suboptimal results. These are probably due to the same gain matrix $\mathbf{G}$ being simultaneously being used by the
> > regularizer and the thresholding mechanism, which tightly couples them together. Hence, both the *weighted consistency regularizer and thresholding mechanism* are together required for the proper functioning of CSST.
> >
> > We would like to mention that in case of maximizing the Worst-Case recall (Sec. 5) the <strong>G</strong> matrix is a *diagonal matrix*, where the proposed thresholding mechanism degenerates to the same thresholding mechanism as of FixMatch. A detailed discussion on it has been added in Appendix Sec. I regarding the same. It can be seen in Table 2 that yet our method CSST(FixMatch) outperforms FixMatch by 24\% in min-recall, hence the proposed weighted consistency regularizer is also an important component in addition to the thresholding mechanism. Also below we compare with FlexMatch (NeurIPS 2021) and DARP (NeurIPS 2020) which use adaptive thresholding mechanisms, where we still find that they still perform inferior to our proposed CSST(FixMatch) model.
> >
> >
> > ### Response to 4.
> > > why only one baseline is being compared to in each task?
> >
> > 1)**Novelty of Task**:  We would like to clarify that to the best of our knowledge there are no existing works that aim to optimize *non-decomposable metrics in a semi-supervised learning (SSL) setup*. We are the **first to propose a framework for the novel objective of optimizing non-decomposable objectives through SSL**, which is an important contribution to the community. There have been some recent works that aim to improve mean recall on imbalanced in a semi-supervised learning paradigm. We provide a comparison of that under similar objectives as ours below (using the official codes):
> >
> >
> > **Table**: Maximising minimum recall for CIFAR10 and minimum of Head and Tail recall for CIFAR-100
> >
> > | Method         |       CIFAR-10  (Imbalance=100) |       |      CIFAR-100  (Imbalance=10)     |      |
> > |:----------------:|:-------------:|:--------------:|:-------------:|:------------------:|
> > |                | Mean Recall | Min Recall | Mean Recall | Min H-T Recall |
> > | CReST[1]          | 0.72        | 0.47       | 0.52          | 0.46           |
> > | DARP[2]           | 0.81        | 0.64       | 0.55          | 0.54           |
> > | FlexMatch[3]           | 0.80        | 0.48       | 0.61          | 0.39           |
> > | CSST(FixMatch)[4] | 0.76        | 0.72       | 0.63          | 0.61           |
> >
> >
> >
> > **Table**: Maximising Mean Recall with a target (tgt.) coverage constraint
> >
> >
> > | Method         |       CIFAR-10  (Imbalance=100) |       |      CIFAR-100  (Imbalance=10)     |      |
> > |:----------------:|:-------------:|:--------------:|:-------------:|:------------------:|
> > |                | Mean Recall | Min Coverage | Mean Recall | Min H-T Coverage|
> > |                |  | (tgt.  0.095) | |  (tgt. 0.01)|
> > | CReST [1]         | 0.72        | 0.052        | 0.52          | 0.009            |
> > | DARP   [2]        | 0.81        | 0.063        | 0.55          | 0.006            |
> > | FlexMatch[3]           | 0.80        | 0.046       | 0.61          | 0.006           |
> > | CSST(FixMatch) [4]| 0.80        | 0.092        | 0.63          | 0.010            |
> >
> > We find that our framework CSST(FixMatch) is able to *improve significantly over even the recent baselines for the objectives of interest* i.e. **min-recall (Table 1)** and **min coverage constraint (Table 2)**. These objectives are of practical importance in areas like fairness where optimizing objectives like worst-case recall by trading off mean recall a bit is common for practical purposes[5,6]. In terms of mean recall, our approach performs better or is on par in 3/4 cases, hence achieving a *better trade-off in terms of satisfying the objective of interest along with the mean metric*.

---

> > > ### Author Response · Authors · 2022-08-03
> > > **Response to Reviewer FQ9c (3/3)**
> > >
> > > 2)**Generality of CSST:** We find that though these above methods in Table improve mean recall, they still are sub-optimal on the particular non-decomposable metrics (i.e., **min-recall (Table 1)** and **min coverage constraint (Table 2)**) we aim to optimize. Hence, this clearly shows the advantage of our proposed CSST framework over other techniques, which are aimed at just improving the mean recall in a general sense. Also, our framework is general and can be plugged into any SSL method which uses a consistency regularizer and thresholding. As FixMatch and UDA are widely used and cited consistency-based methods, we plug them into CSST and show improvements.
> > >
> > > > Discussion on Limitations
> > > - We have discussed Limitations in Appendix A.1 and referred to that in the checklist.
> > >
> > >
> > > References:
> > >
> > > [1]: Chen Wei, Kihyuk Sohn, Clayton Mellina, Alan Yuille, Fan Yang, CReST: A Class-Rebalancing Self-Training Framework for Imbalanced Semi-Supervised Learning, CVPR '21 \
> > > [2]: Jaehyung Kim, Youngbum Hur, Sejun Park, Eunho Yang, Sung Ju Hwang, Jinwoo Shin, Distribution Aligning Refinery of Pseudo-label for Imbalanced Semi-supervised Learning NeurIPS '20 \
> > > [3]: Bowen Zhang, Yidong Wang, Wenxin Hou, Hao Wu, Jindong Wang, Manabu Okumura, Takahiro Shinozaki
> > > ,FlexMatch: Boosting Semi-Supervised Learning with Curriculum Pseudo Labeling NeurIPS '21\
> > > [4]: Cost-Sensitive Self-Training for Optimizing Non-Decomposable Metrics (Ours) \
> > > [5]: Muhammad Bilal Zafar, Isabel Valera, Manuel Gomez Rodriguez, Krishna P. Gummadi, Fairness Constraints: Mechanisms for Fair Classification, AISTATS'17
> > > [6]: Mehryar Mohri, Gary Sivek, Ananda Theertha Suresh,Agnostic Federated Learning, ICML 2019

---

> > > > ### Comment · Reviewer_FQ9c · 2022-08-09
> > > > **Rebuttal**
> > > >
> > > > Thanks for the answers. I think the paper should have been submitted in a readable form in the first submission.
> > > > I want to wish success in the next submission with a better version of the paper that should be further improved both with respect to the writing clarity and the quality of the experiments. Specifically, the notations should made plainly and not ambiguous  and the ablation, especially with respect to the threshold selection should be (much!) more comprehensive.
> > > > The idea in the paper might be of significance to the community but part of making it significant is writing it in a readable form.

---

> ### Author Response · Authors · 2022-08-08
> **Request for Feedback**
>
> Dear Reviewer FQ9c,
> We sincerely thank you for providing helpful feedback on our work, which has significantly improved the quality of our paper. Also, we think that the majority of your initial impression of our work was based on the misunderstanding
> *of learned classifier $\hat{F}$ to be wrongly assumed as optimal classifier $F^\*$* in Theorem 5, which we have clarified in our response. We would be grateful if you could please go over our response and let us know if you have any further concerns.
>
>
> Thanks,
> Authors

---

### Official Review · Reviewer_Vi6Y · 2022-07-17

**Rating:** 4
**Confidence:** 2
**Soundness:** 3 good
**Presentation:** 1 poor
**Contribution:** 2 fair

**Summary:**

The paper proposes a technique for optimizing non-decomposable metrics in self-supervised learning settings. The method is based on the hybrid loss for optimizing non-decomposable metrics in supervised learning settings (Narashiman & Menon, 2021). The authors apply the reduction of the non-decomposable metric to cost-sensitive learning and form a weighted consistency regularizer. The framework is then used to improve the FixMatch algorithm for self-supervised training. Finally, the authors demonstrate the benefit of the models in real-world datasets.

**Questions:**

Please answer my concerns and questions in the previous section.


**Limitations:**

The authors have not discussed the limitation of the proposed model. A discussion on the model's limitations is suggested.

**Strengths And Weaknesses:**

Strengths:
- Optimizing non-decomposable metrics are relevant to many real-world problems
- The authors provide theoretical justifications for the proposed models.
- The experiments show the benefit of the proposed algorithm compared to baselines.

Weaknesses:
- My biggest complaint about the paper is the clarity of the paper. The paper is hard to understand for someone that is not already familiar with the related works. In many places, the authors refer to the related works without providing enough explanation for the reader to understand. In short, I do not think the authors have made the paper self-contained.
- Some terminologies in the paper are used before even defining them, for example, Assumption 1 in Sec 3.2 uses F_pl, but it is defined only after Sec 3.4.

----------- post rebuttal -----------

Thanks to the authors for providing the detailed explanation.
The authors has added more explanation in the appendix.
However, I still think the paper need more explanation on the main paper to add clarity and help reader that is not already familiar with the related works. As also mentioned by another reviewer, the paper is very hard to read.
For the technical contribution of the paper, I could not asses it thoroughly due to the difficulty of understanding the contribution of the paper.

---

> ### Author Response · Authors · 2022-08-02
> **Response to reviewer Vi6Y**
>
> We thank the reviewer for the suggestions and comments. Below we respond to the weaknesses mentioned.
>
> > Clarity of The Work
>
> In the revised version of our paper, we have improved notational consistency, along with moving the complex and intricate details from the paper to the appendix. The following major changes have been made to improve the clarity of our work:
> - We added Table 1: Metrics defined using entries of a confusion matrix.
> - Line 181-183: Intuitive explanation for the non-decreasing nature of the c-expansion function given.
> - We also made the definitions of $F_{pl}$, $\hat{F}$ and $F^\star$ clearer
> - The main Theorem (i.e. Theorem 5) has been clarified followed by an explanation of the same.
>
> Also, we do agree that our paper has mathematical complexity, but this is due to the nature of work on the problem of non-decomposable objectives[20,21,22]. *We have now provided intuitive examples and explanations for improving readability*.
>
> > Reference to earlier work and Self Containment
>
> Thank you for your suggestion. We now formally state the results used from existing works in Appendix N of the revised version. Some further additions to make the paper self-contained have been listed as follows:
> - Section N.1: Bayes optimality of Cost-Sensitive-Loss introduced.
> - Section N.2: Comparision of c-expansion with (a, $\tilde{c}$ )-expansion
> - Section N.3: Assumption of disjoint support assumed in Wei et al[38] as compared to our setting where we do not require the existence of disjoint support.
> - Example 8 (Appendix): c-expansion property shown to hold for a mixture of an isotropic gaussian function
>
> We *sincerely request you to please have a look at the revised version of our paper* and refer to the summary of revisions made to improve the clarity of the paper. Please let us know in case you have any further concerns.

---

### Author Response · Authors · 2022-08-02
**Summary of Revision**

## Summary of Revision

Dear reviewers,
We sincerely thank you for all your feedback. We have posted a revised version of the draft, in which we have made significant efforts to *improve the readability of the draft by providing examples and intuitions for theoretical results*. However, all the theoretical results are still the *same* as the submitted version. We have highlighted the majority of modifications in the draft by font color **blue**.

### List of Modifications/Additions
- Table 1: Metrics defined using entries of a confusion matrix.
-  Line 91: Added the min-max optimization objective of improving worst-case recall using Lagrange multipliers $\mathbf{\lambda}$
- Line 137: Class conditional distributions defined
- Line 156-157: A robust classifier defined in terms of weighted consistency regularization
- Line 159-160: $F_{pl}$ is defined
- Line 164-167:  Assumption 1 elaborated
- Line 176: Domain-Range for c function in c-expansion defined
- Line 180-183: Intuitive explanation for the non-decreasing nature of the c-expansion function given
- Line 302-304: Reference section H in the Appendix regarding the equivalence of confidence-based thresholding to the KL-divergence-based thresholding for diagonal gain matrix and hard pseudo-label.
- Line 586-594(Appendix): c-expansion property shown to hold for a mixture of isotropic gaussian functions
- Section N.1: Bayes optimality of Cost-Sensitive-Loss introduced.
- Section N.2: Comparision of c-expansion with (a, $\tilde{c}$ )-expansion
- Section N.3: Assumption of disjoint support assumed in Wei et al. [38] as compared to our setting where we do not require the existence of disjoint support.
- Section H: Threshold mechanism for diagonal Gain Matrix with hard pseudo-label and its equivalence to simple confidence-based thresholding used in FixMatch.
- Appendix Figure 5: 2 distributions with disjoint supports
- Appendix Figure 6: 2 distributions with non-disjoint supports
- Maximising worst-case recall is mentioned as a separate equation line 72-73
- Line 185-187: Equivalence of c-expansion and (a, $\tilde{c}$ ) expansion briefly introduced
- $P_w$ changed to $\mathcal{P}_w$
- Line 202: Error bound on the learned classifier defined more clearly

Thanks
Authors

---

### Author Response · Authors · 2022-08-08
**General Request for Response**

We thank all the reviewers for their interesting questions and constructive feedback. We have carefully responded to each point raised in the reviews. We hope that the response clarifies all the questions. Please let us know if any further clarifications are required.

---

### Meta-Review · Area_Chair_MYfW · 2022-08-24

**Recommendation:** Accept
**Confidence:** Certain

**Metareview:**

The paper received two negative scores 3/4 (another is 8 with high confidence 5) and the main critcism is that the writing is vague especially for the topic of the paper may not be very popular to the community. The authors have made good efforts in improving their presentation and they also provide additional clarifications as well as new experimental results point to point. Hence in my opinion, the new pdf version is more readable.

For its significance and novelty, the proposed new loss with regularization is theoretically sound and empirically effective. It also addresses the self-training for non-decomposable metrics which is the first time in literature to our knowledge. There are many applications for this method and there are little related work in the community which highlights its potential impact.

I suggest to accept this paper for its significance, quality and strong results. The writing is also improved during the rebuttal.

**Award:**

No

---

### Decision · Program_Chairs · 2022-09-14

Accept